

# Evolution of Anthropogenic Air Pollutant Emissions in Guangdong Province, China, from 2006 to 2015

Yahui Bian[a], Jiamin Ou[b], Zhijiong Huang[c*], Zhuangmin Zhong[a], Yuanqian Xu[a], Zhiwei Zhang[a], Xiao Xiao[a], Xiao Ye[a], Yuqi Wu[a], Xiaohong Yin[a], Liangfu Chen[d], Min Shao[c], Junyu Zheng[a,c,*]

[a] *School of Environment and Energy, South China University of Technology, Guangzhou 510006, China*

[b] *School of International Development, University of East Anglia, Norwich NR4 7TJ, UK*

[c] *Institute for Environmental and Climate Research, Jinan University, Guangzhou 510000, China*

[d] *State Key Laboratory of Remote Sensing Science, Institute of Remote Sensing and Digital Earth, Chinese Academy of Sciences, Beijing 100101, China*

Corresponding author:

Zhijiong Huang

**Phone**: +86-20-37336635

**fax**: +86-20-37336635

**e-mail**: bmmj@163.com

**address**: Institute for Environmental and Climate Research, Jinan University, Guangzhou 510006, China

Junyu Zheng

**Phone**: +86-20-37336635

**fax:** +86-20-37336635

**e-mail**: zhengjunyu_work@hotmail.com

**address**: Institute for Environmental and Climate Research, Jinan University, Guangzhou 510006, China



## Abstract

Guangdong province (GD), one of the most prosperous and populous regions in China, still experiences haze events and growing ozone pollution, although it has seen substantial air quality improvement in recent years. Co-control of fine particulate matter ($PM_{2.5}$) and ozone in GD calls for a systematic review of historical emission patterns. In this study, emission trends, spatial variations, source-contribution variations, and reduction potentials of sulfur dioxide ($SO_2$), nitrogen oxides ($NO_X$), $PM_{2.5}$, inhalable particles ($PM_{10}$), carbon monoxide (CO), ammonia ($NH_3$), and volatile organic compounds (VOCs) in GD from 2006 to 2015 are revealed using a dynamic methodology, taking into account economic development, technology penetration, and emission controls. The relative change rates of anthropogenic emissions in GD during 2006-2015 were -48% for $SO_2$, -0.5% for $NO_X$, -16% for $PM_{2.5}$, -22% for $PM_{10}$, 13% for CO, 3% for $NH_3$, and 13% for VOCs. The declines of $SO_2$, $NO_x$, $PM_{2.5}$, and $PM_{10}$ emissions are mainly attributed to the control-driven emission reductions in the Pearl River Delta (PRD) region, especially from power plants, industrial combustion, on-road mobile sources, and fugitive dust, and partly to the shift of industries from the PRD to the non-PRD (NPRD) region in GD. NPRD also contributed to part of the emission decline, but it was only effective until 2011 when GD's Clean Air Action of *12th Five-Year* was implemented. Due to the growth of solvent use and the absence of effective control measures, VOC emissions in PRD and NPRD both steadily increased, and this might be one of the reasons that led to the slight upward trends of ozone concentrations in GD. To further reduce emissions, future work should focus on power plants and industrial combustion in GD and industrial process source in NPRD for emissions of $SO_2$, $NO_X$, and particulate matter, and on solvent use and on-road mobile sources for VOC emissions. This study provides solid scientific support for further air quality improvement in GD. In addition, it provides robust data to quantify the impact of emission and meteorology variations on air quality and unveil the primary cause of significant air quality change in the PRD region in recent decade.

**Keywords:** emission trends; source contribution; Guangdong Province; emission reductions





# Introduction


China, the world's most prolific emitter of anthropogenic air pollutants, has been
working on ways to curb the deterioration of its air quality in recent decades. After the
launch of the "Clean Air Action Plan" (CAAP) in 2013, China has seen a dramatic
reduction in emissions, mainly driven by control measures in power plants and
industrial sources. In 2017, the Ministry of Environmental Protection declared that
China had achieved the desired targets of CAAP. Average fine particulate matter ($PM_{2.5}$)
concentrations fell 35% in 74 cities across China from 2013 to 2017 (Zheng et al., 2017).
China's Guangdong Province (GD), which is one of the most prosperous and populous
provinces in China (Fig. S1), is one of the regions that has experienced significant air
quality improvement in recent years. Particularly the Pearl River Delta (PRD) region,
known as the hub of the "World Factory," is the first region to meet China's national 35
$\mu g/m^3$ $PM_{2.5}$ standard for three consecutive years (34 $\mu g/m^3$ in 2015, 32 $\mu g/m^3$ in 2016
and 34 $\mu g/m^3$ in 2017).
However, air pollution in GD is still a major concern. First, the annual $PM_{2.5}$ levels
still far exceed stricter air-quality standards, such as the WHO IT-2 (25 $\mu g/m^3$). Also,
haze events frequently occur in winter (Tao et al., 2017). Second, the ambient ozone
concentrations have been growing in recent years, a phenomenon also observed in
northern China. The $90^{th}$ percentile of the maximum 8-hour average ozone
concentration (90%-8h-ozone) in the PRD region was 165 $\mu g/m^3$ in 2017, 14% and 9%
higher, respectively, than those in 2015 and 2016 (GDEMC, EPDHK, EPBMC, 2018).
Further mitigation of air pollution in GD calls for a systematic review of historical
emissions, which could help policymakers understand the evolution of emissions,
quantify the cuts in emissions that have been achieved by control measures, and identify
those sources with the greatest potential for large future emission reductions (Gurjar et
al., 2004; Ohara et al., 2007; Zhong et al., 2013). This is particularly important for those
regions that have less potential for further emission reduction as control measures
tighten. In addition, it is essential to achieve a long-term emission inventory to reveal
the main causes of changes in air quality, a step that is controversial at present. For
instance, Lin et al. (2018) suggested that emission controls helped to improve local air
quality in the PRD region, since there was a high consistency of ambient $PM_{2.5}$
concentrations and emissions in that region. However, Mao et al. (2018) argued that
meteorological and climate conditions rather than PM emissions are in control of the
interannual variabilities and trends of winter haze days in PRD based on an observation-
based approach. Using atmospheric chemical transport models (CTMs) and long-term



emission data, the impacts of emission changes and control measures on air quality can
be quantified. Coupled with other models, the corresponding impacts on climate change
and population exposure can also be assessed. All this information is crucial to guiding
future air-quality management and formulating robust air-quality policies.

Guangdong province was one of the first areas in China to compile emission

inventories (Zhong et al., 2013). The local government has published PRD regional
emission inventories for the base years of 1997, 2001, and 2003 using a top-down
approach to support air-quality management. Zheng et al. (2009) developed the first
high-resolution emission inventory for the PRD region in 2006. This inventory included
six major categories and seven pollutants. Subsequently, emission inventories for black
carbon (BC), organic carbon (OC), ammonia ($NH_3$), and biogenic volatile organic
compounds (VOCs) in PRD were developed for different base years (Yu et al., 2011;
Yin et al., 2015; Li et al., 2016). Huang et al. (2015) expanded the 2006-based emission
inventories in PRD to provincial inventories in GD. Pan et al. (2015) updated the GD
emission inventories by advancing the base year to 2010 and including additional
emission sources. More recently, Zhong et al. (2018) updated the 2012-based GD
emission inventories by source classification, emission methods, emission factors, and
spatial-temporal surrogates. Regarding the emission trend, Lu et al. (2013)
characterized anthropogenic emission trends and their variations in the PRD region
from 2000 to 2009. Liu et al. (2017) developed long-term vehicle emissions in GD from
1994 to 2014.

One limitation of the above-mentioned studies is that most of them were carried

out in a single year or in a limited region, or considering limited sources, so that they
vary in methodology and source classification. Furthermore, most of these studies
focused mainly on the PRD region due its notable economic growth and urbanization,
but ignored the NPRD region that generally had less emissions. However, due to the
strengthened emission controls in PRD and the shift of industries that are energy
intensive, or highly polluting, or have excess production capacity from the PRD region
to non-PRD (NPRD) areas (Chun, 2012; Yin et al., 2017) (Fig. S2), emissions in the
PRD and NPRD regions might have experienced substantial changes in recent years.
Therefore, there is a need to develop a long-term historical emission inventory in GD
using a consistent methodology and the same underlying driver data to fill the data gaps
and to assist with future air pollution control measures.

In this study, we developed a multi-year anthropogenic emission inventory for $SO_2$,

$NO_X$, $PM_{10}$, $PM_{2.5}$, CO, VOCs, and $NH_3$ for the years from 2006 to 2015 using a
dynamic methodology that considers economic development, technological penetration,



and emission controls. The emission trends were validated by ground-based
measurements and satellite observations. Based on the long-term historical inventory,
the emission changes, contribution variations, possible causes and reduction potentials
in 2020 in PRD and NPRD were analyzed and compared, which could provide scientific
evidence for future air quality regulations in GD. Also, the long-term emission
inventories developed in this study are essential data to evaluate the effectiveness of
emission control measures and identify the dominant cause of significant air quality
change in the PRD region.

## 1 Methodology and data

### 1.1 Methods for emission estimations

In this study, we applied a dynamic technology-based methodology that considers
economic development, technological penetration, and emission controls to estimate
the anthropogenic emission trends in GD, following previous studies on emission trends
(Streets et al., 2006; Zhang et al., 2007; Lu et al., 2013). We estimated emissions of 7
pollutants ($SO_2$, $NO_X$, $PM_{10}$, $PM_{2.5}$, VOCs, CO, and $NH_3$) from 13 major categories and
70 sub-categories based on Pan et al. (2015) and the guidelines for the development of
an air-pollutant emission inventory for Chinese cities (MEPC, 2017). The major
categories include power plants, industrial combustion, residential combustion, on-road
mobile, non-road mobile, dust sources, industrial processes, industrial solvent use, non-
industrial solvent use, storage and transportation, agricultural sources, biomass burning,
and other sources (Table S1). Except for on-road mobile and construction dust sources,
emissions of most anthropogenic sources were calculated as follows:
$$E_{i,n} = \sum_{i,j,k} A_{i,j,k,n} \sum_{m} (X_{i,j,k,m,n} EF_{j,k,m,n}) \sum_{z} [C_{z,n}(1 - \eta_z)] \qquad (1)$$

where $i, j, k, m, n,$ and $z$ represent the city, the emission source, the type of fuel or
product, the production technology, the year, and the control technology, respectively,
$A$ represents the activity level (such as the fuel consumption or material production), $X$
represents the percentage of fuel or production for a sector consumed or produced by a
specific technology $m$, $EF$ is the unabated emission factor, $EF \sum_z [C_z(1 - \eta_z)]$ is the
net EF after applying control technology, $C$ is the penetration of the control technology
$z$, $\eta$ is the removal efficiency of the control technology $z$, and $S$ and $SR$ represent the
sulfur content in fuel and the sulfur retention in ash, respectively. For fuel combustion,
the EF of $SO_2$ was estimated using the following equation:
$$EF = 2 \times S \times (1 - SR) \qquad (2)$$



For on-road mobile sources and construction dust, emissions were estimated by
Eqs. (3) and (4), respectively. The methods from previous studies employed for the
other emission sources are listed in detail in Table S2 in the Supplementary Material
(SI).
$$E_{i,n} = \sum_{i,j} (P_{i,j,n} \times VKT_{i,j,n} \times EF_{j,n}) \tag{3}$$

$$E_{i,n} = \sum_{i} (S_{i,n} \times T_{i,n} \times EF_{i,n}) \tag{4}$$

where $i$, $j$, and $n$ represent the city, the vehicle type, and the year, respectively, $P$ is the
vehicle population, and $VKT$ is annual average vehicle kilometers traveled. $i$ and $n$
represent the city and the year, respectively. $S$ is the construction area, and $T$ is the
construction cycle.
The annual emission mainly depends on activity data, emission factors, and
removal efficiencies of emission controls. Therefore, an accurate representation of the
annual change of activity data and emission factors is critical for characterizing the
emission trend. Here, we provided a detailed description of activity data and emission
factors applied in this study.

### 178    1.1.1 Activity data

The estimation of a multiyear emission inventory is complicated since it requires
consistent and accurate activity data and EFs (Zhang et al., 2007). Most of the activity
data during 2006-2015 in this study were obtained from officially released statistics or
relevant reports. Either surrogate data or data interpolation was used to fill in the data
for some sources that lack continuous and consistent long-term activity data. Notably,
the activity data that specifies an individual industry or power plant, defined as point
data, were preferentially used, since these data generally have detailed information on
the location, technical level and control measures. Otherwise, activity data at the city
level, known as areal data, were adopted as a second choice. In this study, for power
plants and industrial combustion we used a combination of point data and areal data to
characterize the activity level, while the other sources all relied on areal data. The
detailed data sources are summarized in Table S2. Here, we describe the processing of
activity data for some major sources, e.g., industrial combustion, construction dust,
marine and on-road mobile sources.
For industrial combustion, we used the total consumption of different energy types
during 2006-2015 from the GD Statistical Yearbook (GDPBS, 2007-2016) to represent
the activity level of each city. Also, we used a detailed dataset from GD pollutant





statistical reports to estimate consumption value of different energy types, the averaged
sulfur contents, and removal efficiencies of industrial combustion in each city. This
dataset, which records the annual fuel consumption, sulfur contents, control devices,
removal efficiencies, product output, and the geographic location of each plant,
contains about 85% of the plants in GD and covers the years of 2006, 2010, 2012, 2014,
and 2015. For the years that lack a detailed dataset, the averaged sulfur contents and
removal efficiencies were estimated by linear interpolation and emission control policy.

For construction dust sources, we used the total annual construction area and

construction cycle time to represent the activity level. The construction area data were
derived from the GD city statistical yearbook (GDPCSY, 2007-2016), and the
construction cycle time was determined by the time requirement for different
construction phases, i.e., earthwork excavation, foundations, earthwork backfill, and
general construction. Considering the effect of rainfall in suppressing dust sources, we
revised the construction cycle time by combining our previous study (Yang, 2014) with
rainfall information for each year obtained from the GD Meteorological Service
(GDMS, 2007-2016).

Regarding marine sources, the characterization of activity level was based on

heavy and diesel fuel consumption. However, it is challenging to acquire detailed
consumption of various fuel types directly. Thus, we used the method described in Li
et al. (2017) to convert the cargo volumes and transport distances of major navigation
routes into fuel consumption data via fuel consumption rates. Fuel consumption rates
were taken from the IMO report (IMO, 2009). The cargo volumes in each city were
obtained from GD Statistical Yearbooks (GDPBS, 2007-2016). Transport distances of
major navigation route data were measured by the historical AIS-based digital map.

For on-road mobile sources, population data of different vehicle types (i.e.,

passenger trucks, buses, taxis, and motorcycles), the gross weight (heavy and light duty)
from the statistical yearbooks, and annual average vehicle kilometers traveled from a
field survey of some cities in GD were used to characterize the activity level. We further
differentiated diesel and gasoline vehicles to obtain a more accurate estimate, but this
information was not available from the official statistics. Therefore, we distinguished
the vehicle population based on test results regarding the vehicular ratios of fuel use in
GD (Che et al., 2009) following a method in our previous study (Lu et al., 2013).

### 1.1.2 Emission factors (EFs)

EFs could have changed with the implementation of emission controls in GD

during 2006-2015 (Table S3), which involve technological penetration and evolution.



To deal with that possibility, we developed a dynamic method to reflect the response of
EFs to control measures and technological penetration. First, we established the
unabated EF of each source to represent what the emission level would have been
without any treatment. The unabated EFs for most emission sources of various
pollutants (i.e., $NO_X$, $PM_{10}$, $PM_{2.5}$, VOCs, CO, and $NH_3$) used in this study are listed in
Tables S4-9, and are based mainly on the latest research results and values
recommended in related manuals of air-pollutant emission coefficients. Next, we
estimated the net EFs of each source according to the corresponding processing
technologies, control technologies, and removal efficiencies that might vary with years.
In this study, we applied the dynamic method to all emission sources. In the following
subsection, we mainly describe their application to major sources that have received
intensive control measures in the past decade: including power plants, industrial
combustion, VOCs-related sources, and on-road mobile sources.

The net EFs of $SO_2$ for industries and power plants in GD were determined based
on removal efficiencies and fuel sulfur content. The annual removal efficiencies and
sulfur content in 2006, 2010, 2012, 2014, and 2015 were obtained from GD pollutant
statistical reports (GDPSR, 2006, 2010, 2012, 2014, 2015) for industries; those
parameters for power plants in 2006-2014 were obtained from power-plant reports
(CSPG, 2006-2014). For years without these documented data, an interpolation method
that considers newly released regulations of $SO_2$ emission controls and expert judgment
were used to estimate the removal efficiency and sulfur contents (2007-2009, 2011, and
2013 for industries and 2015 for power plants). For instance, the sulfur content of coal
and oil in industrial sources can be estimated as <0.7% and 0.8%, respectively,
according to the *Guangdong industrial boiler pollution remediation program (2012-*
*2015)* released in 2012.

For VOCs-related sources, such as industrial solvent use, non-industrial solvent
use, and industrial process sources, the net EFs of VOCs were determined based on the
installation rate of VOC control technologies and the removal efficiencies, which were
acquired by an on-site investigation of VOCs-related industries in GD. Additionally,
the new VOC emission standards were also used to determine in which year VOC
control technologies were implemented. For example, emission standards for furniture
surface coating and shoemaking were implemented in 2010 (*Emission standard of*
*volatile organic compounds for furniture manufacturing operations (DB44/814-2010),*
*Emission standard of volatile organic compounds for shoe-making industry*
*(DB44/817-2010))*. Thus, we estimated the net VOC EFs for furniture surface coating
and shoemaking with the consideration of VOC removal efficacies since 2010. For the




vehicular EFs, we used the same method employed in our previous study (Lu et al.,
2013). The vehicular EFs were calculated based on the 2007 International Vehicle
Emissions (IVE) model (ISSRC, 2008), while the EFs for other years were derived from
2007-based EFs in consideration of emission standards, fuel standards, and vehicle
lifespans.
**1.2 Data for validation**

To validate the multi-year emissions in GD, we compared the emission trends with
satellite-based data. The $SO_2$ column amount (OMSO2e v003) and the $NO_2$
tropospheric column (OMNO2d v003) were retrieved from the Ozone Monitoring
Instrument (OMI) with a spatial resolution of 0.25°×0.25° (available at
https://giovanni.gsfc.nasa.gov/giovanni/). Aerosol optical depth (AOD) data were taken
from the Moderate-Resolution Imaging Spectroradiometer (MODIS) aerosol product
MOD04 with a high resolution of 10 km (available at
https://ladsweb.modaps.eosdis.nasa.gov/). In addition, ground-level observations
obtained from the PRD air-quality monitoring network (GDEMC, EPDHK, EPBMC,
2007-2016) were also used to validate emission trends in PRD. The air-quality
monitoring network came into operation at the end of 2005 and has provided accurate
air quality data to local governments and the public. These data from the PRD air quality
monitoring network were adopted because of the high reliability of the operating
procedures on quality assurance and quality controls.

## 2 Results and discussion

### 2.1 Overall emission trends

The overall emission trends of $SO_2$, $NO_X$, $PM_{10}$, $PM_{2.5}$, VOCs, CO, and $NH_3$ in
GD during 2006-2015 are presented in Fig. 1. From 2006 to 2015, anthropogenic
emissions decreased by 48% for $SO_2$, 0.5% for $NO_X$, 16% for $PM_{2.5}$, and 22% for $PM_{10}$,
but increased for CO, $NH_3$, and VOCs, by 13%, 3%, and 33%, respectively. Specifically,
$SO_2$ emissions fell steadily during 2006-2015, which might be due to the strict controls
on $SO_2$ emissions implemented in the 11th Five Year Plan (FYP) (2006-2011). $NO_X$
emissions overall showed an upward trend in the early period, reaching a peak in 2011.
After the implementation of *the Planning for Guangdong province environmental protection*
*and ecological construction in 12th FYP* (PGGP, 2011) in 2011, in which $NO_X$ emission
caps of all industrial sectors were proposed, $NO_X$ emissions decreased, declining by 9%
in 2015. The $PM_{10}$ and $PM_{2.5}$ emissions showed an increasing trend during 2006-2009
but then decreased steadily. Similarly, CO emissions showed a small rise during 2006-

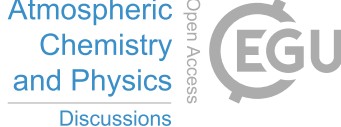



2013, followed by a sharp decline. $NH_3$ emissions changed a little, while VOC
emissions steadily increased over the 2006-2015 period, mainly fueled by the absence
of effective emission-control measures.

Although emissions of $SO_2$, $NO_X$, PM, and CO declined in recent years, the per

capita GDP, fuel consumption, and vehicle population in GD, which account for most
anthropogenic pollution activity, saw growth from 2006 to 2015, as shown in Fig. 1.
From 2006 to 2015, the per capita GDP and vehicle population significantly increased,
by 135% and 66%, respectively. Obviously, anthropogenic emissions in GD were
decoupling from economic and energy consumption growth. This means that the
emission regulations and control measures enacted in GD have alleviated emissions
despite the growth of economic activity. For instance, $NO_X$ emissions are closely related
to fuel consumptions because a large proportion of their emissions are from industries
or power plants that consume a great deal of fuel. However, the trends of $NO_X$ emissions
and fuel consumption have deviated from each other since 2011 when low $NO_X$-
combustion (LNB) control measures and flue gas denitrification technology (i.e.,
selective catalytic reduction (SCR) and selective non-catalytic reduction (SNCR)) in
power plants were enacted.

We have also compared emission trends in PRD and NPRD (Fig. S3). Since

emissions in GD were mainly concentrated in the PRD region, emission trends in PRD
were similar to those in GD, except $NO_X$ emissions, which started off steady until 2012
and then fell slightly. Compared with 2006, the 2015 emissions of $SO_2$, $NO_X$, $PM_{2.5}$,
and $PM_{10}$ in PRD decreased by 63%, 14%, 35%, and 27%, respectively, while CO and
VOCs emissions increased by 2% and 35%, respectively. In NPRD, $SO_2$, $NO_X$, $PM_{2.5}$,
and $PM_{10}$ emission trends differed from those in PRD. Compared with 2006, the 2015
emissions of $SO_2$, $PM_{2.5}$, and $PM_{10}$ in NPRD decreased by only 8%, 8%, and 5%,
respectively, but emissions of VOCs, $NH_3$, CO, and $NO_X$ significantly increased—by
30%, 10%, 31%, and 29%, respectively. The discrepancy of emission trends between
PRD and NPRD was expected. This is because these two regions achieved different
levels of progress on air quality management in the past decade due to the discrepancy
in economic development. Overall, most previous control strategies (Table S3) focused
mainly on PRD while the NPRD region received little attention. Moreover, initiated by
the policies of "vacate the cage and change birds" (in Chinese, Teng Long Huan Niao)
initiated by the Guangdong provincial government in 2008 (Li & Fung, 2008; Yang,
2012), many low-tech industries in PRD were relocated to NPRD. Detailed emission
evolutions and the corresponding causes are discussed in Sec. 2.3.





## 2.2 Validation of emission trends


In this section, we validate emission trends in PRD using ground observations and
satellite data (Fig. 2). In general, these two data sources are consistent for emission
trends of $SO_2$, $NO_X$, and $PM_{10}$. During 2006-2015, $SO_2$ emissions in PRD decreased by
63% (emission trends in PRD and NPRD are shown in Fig. 2), slightly less than the 68%
and 86% observed in ground-level and satellite data, respectively. NOx emissions and
observations all showed a declining trend during 2006-2015. For $PM_{10}$ emissions, the
declining trend also closely followed the fluctuant downward trend of ambient $PM_{2.5}$
concentrations and AOD. The fluctuations of observations were associated with $PM_{2.5}$
formation influenced by annual variations of meteorology.
The change in spatial variation of emissions in GD was also evaluated using
satellite measurements. Here, the column concentrations of $SO_2$, $NO_2$, and AOD for
GD in the years 2007, 2011, and 2015 were applied, as illustrated in Fig. 3 (the column
concentrations in the remaining years are displayed in Fig. S4). Overall, the spatial
patterns of the satellite measurements also reveal the different emission trends between
PRD and NPRD. For example, $SO_2$ column concentration in PRD decreased by 71%
from 2007 to 2011, but in NPRD increased by 26%. This agrees with the emission
trends, in which $SO_2$ emissions in PRD decreased by 39% while those in NPRD
increased by 15%. From 2011 to 2015, $SO_2$ column concentration decreased by 31%
and 42% in PRD and NPRD, respectively, and $SO_2$ emissions decreased by 32% and
20% in PRD and NPRD, respectively. From 2007 to 2011, $NO_2$ column concentration
decreased by 16% in PRD but increased by 16% in NPRD. These trends also coincided
with emission changes, which $NO_X$ emissions in PRD decreased by 13% but increased
by 36% in NPRD. From 2011 to 2015, $NO_2$ column concentration and $NO_X$ emissions
both decreased in PRD and NPRD. AOD displayed a decrease of approximately 23%
in PRD and NPRD from 2007 to 2015. A similar pattern was also found in $PM_{2.5}$
emission trends, with decreases of 35% in PRD and 9% in NPRD.
Overall, the above-mentioned validations using satellite observations and ground
measurements demonstrate that emission trends of $SO_2$, $NO_X$, and PM estimated in this
study are reliable.

## 2.3 Evolution of source emissions in Guangdong Province

To further understand the evolution of emissions in GD, we estimated the changes
of anthropogenic emissions by emission category in PRD and NPRD during 2006-2015.
Here, we examine the PRD and NPRD regions separately since these two regions may
have experienced diverse emission evolutions due to their different levels of progress





on air-quality management. The annual source contributions during 2006-2015 are also
presented in Fig. 4 (GD) and Fig. S5 (PRD and NPRD).

### 2.3.1 SO$_2$

In PRD, SO$_2$ emissions steadily declined, from 0.788 Tg in 2006 to 0.292 Tg in

2015. The decline was dominated by power plants and industrial combustion, which
decreased by 0.247 Tg and 0.232 Tg, respectively, in this period (Fig. 5a). This indicates
that flue-gas desulfurization, the control of sulfur content of fuels, and boiler renovation
of SO$_2$ emission controls in the 11$^{th}$ FYP were effective. In NPRD, on the contrary, SO$_2$
emission increased until 2010, when it saw a downturn. Before 2010, SO$_2$ emission
growth was mainly associated with the increase of industrial combustion and non-road
mobile sources (Fig. 5b). These two sources still maintained a slight rise after 2010, but
their increased emissions were offset by a sharp plunge of emissions from power plants,
which resulted mainly from tightening desulfurization technology. To reveal the reason
for the growth of industrial combustion in NPRD, we compared the fuel-consumption
trends from industrial combustion in PRD and NPRD. As shown in Fig. 6, the fuel
consumption from industrial combustion in PRD dropped by 47% during 2006-2015.
By contrast, in NPRD, it increased by 99%, mainly after 2010. This might be closely
associated with the policy of "vacate the cage and change birds" that brought many
energy-intensive industries from PRD to NPRD.

The source contribution of SO$_2$ emission in GD also changed (Fig.5a). In GD, the

contribution of power plants dropped from 43% to 27%. By contrast, the contribution
of industrial combustion remained stable, in the range of 36% to 41%. Therefore,
industrial combustion took over from power plants to became the largest SO$_2$ emission
source in GD. The PRD and NPRD regions also had similar changes (Fig. S5a-b). Note
that the contribution of non-road mobile sources to SO$_2$ emissions increased both in
PRD (from 13% to 29%) and NPRD (from 8% to 17%), due to the absence of effective
emission control measures. Therefore, future work should be focused on non-road
mobile source to further reduce SO$_2$ emissions in GD. In addition, in NPRD, industrial
combustion also might have potential for reduction, since its contribution slightly
increased, from 5% to 11%.

### 2.3.2 NO$_X$

As with SO$_2$ emissions, the evolution of NO$_X$ emissions also differed between

PRD and NPRD. In PRD, NO$_X$ emissions dropped overall from 0.926 Tg in 2006 to
0.797 Tg in 2015, especially after 2012 when NO$_X$ emissions saw a noticeable





downturn (Fig. 5c). Power plants were the principal sources leading to emission
reduction. In response to the explicit target of reducing $NO_X$ emissions in the 12th FYP
(2011-2015), a large number of power plants had installed $NO_X$ removal equipment
since 2011. Thus, in 2015, $NO_X$ emissions from power plants were 0.214 Tg less than
in 2006. However, a part of the $NO_X$ emission reductions from power plants was still
canceled out by the growth in emission from non-road mobile sources, an increase of
0.452 Tg; and on-road mobile sources, with a slight increase of 0.012 Tg. Recent control
measures have focused on vehicle exhaust, but it is unlikely to offset the increasing
vehicle population in PRD, which increased by 88% during 2006-2015. Consequently,
on-road mobile sources (growing from 32% in 2006 to 38% in 2015) overtook power
plants (growing from 43% to 23%) as the largest contributor to $NO_X$ emissions in PRD
since 2010 (Fig. S5c).
Unlike in PRD, $NO_X$ emissions in NPRD rebounded in 2008 and grew sharply
until 2012. The emission changes were mainly from industrial combustion, non-road
mobile sources, and power plants (Fig. 5d). Among these three sources, $NO_X$ emissions
from industrial combustion and non-road mobile sources both showed an upward trend,
increasing by 159% and 84%, respectively, during 2006-2015. These trends can be
explained by the shift of energy-intensive industries from PRD to NPRD and the
absence of catch-up emission controls for industries and non-road mobile sources.
Since 2011, possibly due to the implementation of denitrification technology, $NO_X$
emissions from power plants steadily went down and offset the slight increase in
emissions from industrial combustion and non-road mobile sources. Consequently, the
contribution of power plants to the total NOx emissions in NPRD, which was once a
large contributor, decreased from 44% in 2006 to 28% in 2015. By contrast,
contributions of industrial combustion and non-road mobile sources increased from 14%
to 29% and from 16% to 23%, respectively (Fig. S5d).
On-road mobile was also a major contributor to $NO_X$ emissions in GD (Fig. 5b).
Although the total $NO_X$ emissions of on-road mobile sources changed slightly in 2006-
2015, their sectoral contribution showed significant change, especially in PRD (Fig.
S5c). Here, we further analyze the trends of $NO_X$ emissions from on-road mobile
sources and the vehicle population from 2006 to 2015 for PRD (Fig. 7a) and NPRD
(Fig. 7b). Overall, on-road mobile NOx emissions in PRD were approximately three
times higher than those in NPRD, but their trends were similar. In both PRD and NPRD,
$NO_X$ emissions from heavy-duty diesel truck (HDDT) slightly decreased although the
HDDT population showed growth before 2014. Apparently, this was closely related to
improvement in vehicular emission and fuel standards. $NO_X$ emissions from heavy-



duty diesel vehicle (HDDV) also dropped, partly due to reduced HDDV population.
Unlike the HDDT and HDDV, the population of light-duty gasoline vehicle (LDGV)
increased by a factor of 5 and 6 in PRD and NPRD, respectively. The upsurge of the
LDGV population outpaced the new standards on vehicle emission enacted since 2009
(PGGP, 2009), leading to growth of $NO_X$ emissions from LGDV. Consequently, the
contribution of LGDV to $NO_X$ emissions in PRD surged from 12% in 2006 to 34% in
2015.

Based on the above analysis, it can be concluded that the $NO_X$ emission trend in
GD is dominated by the decline of power plants in PRD and the increase of non-road
mobile source and industrial combustion in PRD and NPRD. Particularly, the
contribution of non-road mobile source and industrial combustion to NOx emissions in
GD increased from 13% in 2006 to 23% in 2015 and from 12% to 20%, respectively
(Fig. 4b), indicating that these two sources should receive more attention in future
emission-control measures, especially industrial combustion in NPRD. Regarding on-
road mobile sources, the largest contributor to NOx emissions in PRD, LDGVs, should
require more attention in the future.

### 2.3.3 $PM_{10}/PM_{2.5}$

As shown in Fig. 4c-d, the main sectoral changes between $PM_{10}$ and $PM_{2.5}$
emission were somewhat similar to each other. Therefore, we focused mainly on the
analysis of $PM_{10}$ emissions. PRD and NPRD showed similar $PM_{10}$ emission variations
regarding trends and source contributions. They both topped out in 2009 and then
decreased monotonically. Compared with 2006, $PM_{10}$ emissions in 2015 dropped by
27% and 6% in PRD and NPRD, respectively. Dust sources, power plants, and
industrial process sources were the major contributors to the change of $PM_{10}$ emissions
in PRD and NPRD. However, emission trends of these three sources in PRD and NPRD
were slightly different, particularly the industrial processes. In PRD, $PM_{10}$ emissions
from industrial processes steadily declined after 2010, while in NPRD, $PM_{10}$ emissions
from industrial processes kept an upward trend during 2006-2015. One possible reason
for the difference is that control measures for $PM_{10}$ emissions in PRD were stricter than
those in NPRD. In PRD and NPRD, fugitive dust increased during 2006-2010 and
showed a decrease during 2011-2015. The downturn in the later years was due to the
implementation of emission control technologies of dust sources in response to the
release of the *clean air action plan for the Pearl River Delta in Guangdong province* in
2010.

As emissions from industrial processes, dust sources, and power plants changed





dramatically, the major sources contributed to $PM_{2.5}$ and $PM_{10}$ emissions also changed
accordingly (Fig. 4c-d, S5g-j). For $PM_{2.5}$ emissions in GD, contributions from dust
sources and power plants declined slightly, from 17% in 2006 to 11% in 2015, and from
12% to 7%, respectively. The contribution from industrial processes, the largest
contributor, also slightly decreased from 38% to 33%. For $PM_{10}$ emissions in GD, the
contribution of industrial processes increased, from 28% to 34%. Particularly in NPRD,
it replaced dust sources as the largest contributor since 2012. In PRD, fugitive dust was
still the largest $PM_{10}$ emission source. Based on the above-mentioned analysis, PM
emission controls for dust sources, especially in PRD, and for industrial processes,
especially in NPRD, should be a priority of the agenda in the next stage of emission
controls. Also, $PM_{2.5}$ emissions from cooking cannot be neglected in GD; this is because
the removal of cooking oil fumes from homes and restaurants was not strictly enforced,
although some regulations and emission standards regarding cooking emissions were
enacted gradually.

### 498     2.3.4 VOCs

As shown in Fig. 5i-j, the sectoral changes of VOC emissions in PRD and NPRD
were similar. The total emissions in these two regions both showed a rising trend during
2006-2015, increasing by 35% and 30% in PRD and NPRD, respectively. The steady
increase mainly originated from the growth of industrial-solvent use and non-industrial
solvent use, whose emissions in GD respectively increased by 99% and 69% during
2006-2015 (Fig. 4e). Industrial solvent use was a large increasing source, especially in
PRD where most industrial sources are concentrated. This is expected because solvent
use required by industrial sources was growing, but control measures were insufficient.
Several VOC control technologies had been adopted since 2010. For instance, the use
of low VOC-containing raw materials for printing, shoemaking, furniture
manufacturing, and other industries was first proposed at the *clean air action plan for the*
*Pearl River Delta in Guangdong province* in 2010. Although these measures slowed an
increasing trend of VOC emissions in PRD (VOC emissions from industrial solvent use
in PRD increased by 18% during 2006-2010 while they increased by 6% during 2011-
2015 (Fig. S6)), the control efficiencies were still low. According to a field survey, only
40% of VOC-emitting industries had removal equipment in 2014 (Wang et al., 2018).
The increasing VOC emissions from industrial-solvent use made it become the largest
contributor to VOC emissions in GD in 2015 (Fig. 4e), with a percentage of 32%.
Therefore, the implementation of policy and upgrade of control technologies are still
required to reduce VOC emissions, especially in PRD. In NPRD, non-industrial solvent



use was also a major contributor to the increase of VOC emissions. In particular, it
became the largest contributor to VOC emissions in NPRD in 2015, with a percentage
of 22%, slightly larger than on-road mobile sources (21%) (Fig. S5j).
Since on-road mobile sources were also a major contributor to VOC emissions,
the evolution of their VOC emissions is also discussed here (Fig. 8). In both PRD and
NPRD, VOC emissions from motorcycles, the largest contributors to VOC emissions
from on-road mobile, went down in the past decade due to the relatively strict ban on
motorcycles. They decreased by 55% and 38% in PRD and NPRD, respectively. By
contrast, VOC emissions from LDGV increased by 118% and 197% in PRD and NPRD,
respectively, likely due to the upsurge of the LDGV vehicle population. Particularly in
PRD, LDGV's became the largest contributor to vehicle-related VOC emissions since
2008, which might also happen in NPRD according to the current trend.

## 2.3.5 CO/NH$_3$

As shown in Fig. 5k-l, CO emissions in PRD and NPRD both increased steadily
during 2006-2013, and then decreased after 2013. However, the sectoral changes were
different in these two regions. In PRD, the growth of CO emissions during 2006-2013
is mainly attributed to industrial combustion and on-road mobile sources, while in
NPRD, it is associated principally with industrial combustion and industrial processes.
The difference exists because on-road mobile sources were primarily concentrated in
PRD while iron and steel sectors, the largest CO emitters among industrial process
sources, were located mainly in NPRD. Notably, production of the iron and steel sectors
soared during 2006-2015, increasing by almost 95% in GD (GDPBS, 2007-2016), but
emission controls fell behind. As to the decline of CO emissions during 2013-2015 in
PRD, on-road mobile was the major reason. By contrast, the slight downturn in NPRD
was mostly due to declining emissions from on-road mobile sources and biomass
burning. All these sectoral changes made industrial combustion (35% in 2015) become
the largest contributor to CO emissions in GD (Fig. 4f). In NPRD, the contribution of
industrial process sources also increased. In contrast, the contribution of on-road mobile
to CO emissions in NPRD decreased by 19% in 2015 compared with that in 2006.
As shown in Fig. 4g, agricultural sources constituted most to the change of NH$_3$
emissions, as they accounted for 86%-87% of the total NH$_3$ emissions in GD. However,
their annual changes were different in PRD and NPRD (Fig. 5m-n). In NPRD, NH$_3$
emissions by agricultural sources increased by 11% during 2006-2015, partly as the
result of growth of fertilizing and livestock to meet the increasing demand for food.
Another reason was the absence of effective emission controls on agricultural sources





in China. By contrast, in PRD, NH₃ emissions by agricultural sources remained stable.

## 2.4 Evaluation of emission control measures and policy implications

Emission changes can be divided into two categories: (1) changes resulting from
change of activity level (activity-driven emission) in the absence of control measures
and (2) changes due to the implementation of pollution controls (control-driven
emission reduction) (Zheng et al., 2018). In this study, these two categories of emission
changes were quantified to help evaluate the efficiencies of the control measures enacted
in GD and to provide implications for future policies. We estimated the unabated
emissions if pollution control had been frozen at the 2006 level. In other words, we
assumed that there were no new control measures adopted since 2006. Then the control-
driven emission reduction was estimated by comparing the unabated emissions and the
actual emissions, and the activity-driven emission was estimated by calculating the
annual changes of unabated emissions. Also, we projected the actual emission to 2020
to help understand the potential for more emission control. Here, the planned emission
controls before 2020 were assumed to be completely implemented in 2020. These
related regulations for $SO_2$, $NO_X$, $PM_{10}$, and VOC emission controls are summarized in
Table S3. The control-driven and activity-driven emissions of $SO_2$, $NO_X$, $PM_{10}$, and
VOCs in 2007, 2009, 2011, 2013, and 2015 in addition to the predicted emissions in
2020 in PRD and NPRD are presented in Fig. 9.

### 2.4.1 SO₂

During 2007-2011, the decline of $SO_2$ emissions in PRD was driven by emission
controls. The control-driven $SO_2$ emission reductions in PRD dramatically grew from
0.120 Tg in 2007 to 0.568 Tg in 2011 (Fig. 9a), mainly attributable to stringent $SO_2$
emission control regulations on industrial combustion and power plants—e.g., shutting
down small coal-fired thermal power units, phasing out small boilers, installing flue-
gas desulphurization (FDG) equipment and limiting the sulfur contents of fuel. $SO_2$
emission reductions from industrial combustion and power plants account for 57%-75%
and 25%-37% of the total emission reductions, respectively. The activity-driven
emissions increased by 0.178 Tg during 2007-2011, but their increments were far less
than the control-driven emission reductions. The control-driven emission reductions
flattened out in recent years. During 2013-2015, the control-driven emission reductions
only increased by 4%, which owes much to the effectiveness of $SO_2$ emission controls
in earlier years and the shrinking of control measures in recent years. Nonetheless, $SO_2$
emissions in PRD still steadily declined, partly due to the decrease of activity-driven



emissions. By contrast, activity-driven emissions in NPRD kept rising, which might be
associated with the transfer of energy-intensive industries from PRD to NPRD (Fig.
9b). Even so, the control-driven emission reductions dramatically increased and
outweighed the activity-driven emissions since 2011, when stricter control measures
were implemented. Similar to the situation in PRD, pollution control-driven emission
reductions in NPRD were mainly attributed to industrial combustion and power plants.
Although $SO_2$ emissions dramatically decreased since 2006, there is potential for
further reduction in 2020. On the basis of control-driven emission reductions in 2015,
$SO_2$ emission reduction potentials in PRD and NPRD in 2020 are projected to be 0.10
Tg (34% of the total $SO_2$ emissions in 2015) and 0.29 Tg (approximately equal to the
total $SO_2$ emissions in 2015), respectively. These reductions can be achieved by
technical innovations, including ultra-low-emission measures in power plants, a series
of actions regarding boiler management, sulfur content controls in fuels, and flue-gas
desulfurization in industries. Most of these emission reductions are from industrial
combustion and power plants. Particularly for NPRD, 60% of the reductions can come
from industrial combustion, more than that in PRD. This is because $SO_2$ removal
efficiencies in industries are still low in NPRD. $SO_2$ emission contribution from non-
road mobile sources in GD previously presented an increasing trend (Fig. 4a). This
reminds us that non-road mobile sources still have a high potential for $SO_2$ emission
reduction. In PRD, it could account for approximately 20% of the total $SO_2$ emission
reductions in 2020. Thus, future measures should be focused on industrial combustion
and non-road mobile sources for controlling $SO_2$ and $NO_X$ emissions ($NO_X$ also
presented a similar result in the subsequent analysis).

**2.4.2 NO_X**
In PRD, the decline of $NO_X$ emissions was driven by emission controls, and was
significantly enhanced in 2011 when the 11[th] Five Year Plan was enacted in GD,
including the application of technology for flue-gas denitrification and low $NO_X$
combustion (LNB) in industries and power plants (Kurokawa et al., 2013), the
elimination of yellow-label cars, and progressive advancements in vehicle emission and
fuel standards. These mitigation measures yielded 0.315 Tg $NO_X$ emission reductions
in 2011, and offset the growth of activity-driven emissions. During 2007-2015, power
plants and on-road mobile sources were the two major contributors, accounting for
34%-59% and 38%-60% of the total control-driven emission reductions, respectively.
Industrial combustion also contributed 2%-14% of the total control-driven emission
reductions. Unlike in PRD, NPRD's new clean-air actions only focused on power plants,



and these measures were not stringent enough to cover the growth of $NO_X$ emissions
before 2011. After 2011, the control measures of $NO_X$ emissions from power plants in
NPRD were strengthened, leading to a significant increase of control-driven $NO_X$
emission reductions. Consequently, the total $NO_X$ emissions slightly declined. Apart
from power plants, on-road mobile sources and industrial combustion also partly
contributed to the control-driven emission reductions in NPRD in recent years.
On the basis of control-driven emission reductions in 2015, $NO_X$ emissions in
2020 could be further reduced by 0.24 Tg both in PRD (30% of the total $NO_X$ emissions
in 2015) and NPRD (43% of the total $NO_X$ emissions in 2015). Most of these projected
reductions could come from industrial combustion and power plants as a result of the
implement of stricter regulations, e.g., ultra-low emissions for power plants, boiler
management, and flue-gas denitrification for industries. In PRD, on-road and non-road
mobile sources also have relatively high potential for $NO_X$ emission reduction.
Particularly, on-road mobile sources, especially LDGV, require more effective control
measures. Although current control measures have alleviated the amount of vehicle
emissions in recent decades, it still cannot cover the increased emissions driven by the
rapid growth of vehicle population, as shown in Fig. 5b. Further reduction of on-road
$NO_X$ emissions can be achieved by the implementation of control regulations.

### 646 2.4.3 $PM_{10}$

The activity-driven emission of $PM_{10}$ in PRD and NPRD both steadily increased
during 2006-2015 due to growth in activity. From 2006 to 2015, the gross industrial
output value increased by 48% (GDPBS, 2007-2016), but control measures were not
implemented until 2009, leading to a slight increase of $PM_{10}$ emissions during 2006-
2009. After 2009, the installation of dust removal equipment dramatically increased
with the stricter implement of PM control measures, such as special requirements
limiting soot emission in power plants, boiler management with smaller capacity, and
a series of pollution controls for non-metallic minerals industries. These measures boost
the control-driven emission reductions, which can counterbalance the growth of
activity-driven emission of $PM_{10}$ in PRD and NPRD. In PRD, the control-driven
emission reductions dramatically improved, from 0.024 Tg in 2009 to 0.318 Tg in 2015,
while in NPRD, they improved from 0.014 Tg in 2009 to 0.258 Tg in 2015. In both the
PRD and NPRD region, industrial combustion, power plants, and dust sources were the
three major contributors to the control-driven emissions reductions.
Compared with control-driven emission reductions in 2015, $PM_{10}$ emissions in
2020 in the PRD and NPRD could be further reduced by 0.31 Tg (60% of the total $PM_{10}$





emissions in 2015) and 0.33 Tg (49% of the total $PM_{10}$ emissions in 2015), respectively.
Fugitive dust is the most significant contributor, accounting 45% and 31% of the total
reductions in PRD and NPRD, respectively. This can be achieved by applying online
monitoring technology for supervising construction dust (Sun et al., 2016) and more
advanced measures, such as achieving a "6 100%" target for construction sites and
increasing machine cleaning ratio for road dust. Industrial process sources, power plants,
and industrial combustion also have major potential to achieve reduction in
emissions—especially industrial process sources in the NPRD.
### 672 2.4.3 VOCs
For VOCs, control-driven emission reductions in PRD and NPRD were slight in
the past decade. Although VOC emission-control measures, such as promoting emphasis
on strict end-of-pipe controls and leak detection and repair (LDAR) technology in VOC-
emitting industries, and strengthening oil and gas recovery in gas stations, have been
gradually highlighted since 2014 when *action plan for air pollution prevention and*
*control in Guangdong province (2014-2017)* (GDEP, 2014) was released, the regulation
has not been well executed. Emission reductions from solvent sources, the largest
contributor to VOC emissions in PRD (Fig. S5i), were 0.075 Tg in 2015, highly
associated with the use of low-VOC products and environmental-friendly paints that
contain low or even no VOCs. However, these emission reductions only accounted for
10%-35% of the total VOC emission reductions. In fact, 65%-86% of the control-driven
VOC emission reductions were from on-road mobile sources, which is mainly
attributed to the improvement of emission standards and oil quality for vehicles,
management of yellow label cars, and the popularization of green traffic. Even so, the
control-driven VOC emission reductions (from 0.016 Tg in 2007 to 0.294 Tg in 2015)
were far outweighed by the activity-driven growth in emissions (from 0.699 Tg in 2007
to 1.172 Tg in 2015), resulting from the growth of vehicle populations and increasing
use of solvents, which consequently drove up VOC emissions in PRD and NPRD (Fig.
5i-j).
In 2020, if the existing emission control regulations were fully implemented, VOC
emissions in PRD would decrease by 30% relative to the emission level in 2015. The
emission reduction potentials are 0.61 Tg in PRD (69% of the total VOC emissions in
2015) and 0.56 Tg in NPRD (0.2 times higher than total VOC emissions in 2015),
respectively, much larger than the emission reduction potentials of $SO_2$, $NO_X$, and $PM_{10}$.
Reduced solvent use is the largest factor. This is because current VOC end-pipe removal
efficiency in GD is still low. Therefore, VOC emissions from solvent use could be



greatly reduced by improving end-pipe removal efficiency. In fact, VOC emission
controls on solvent use and industrial process source were particularly prioritized
during the 13th FYP (2016-2020). If the VOCs end-pipe removal efficiencies achieve
their control targets in the 13th FYP, VOCs emission reductions from solvent use will
be 0.388 Tg and 0.257 Tg in PRD and NPRD, respectively, accounting for 43% and 38%
of the total VOC emission reductions. Another source with large potential for emission
is on-road mobile sources.

## 707    3 Summary and conclusions

In this study, we provide a detailed examination of anthropogenic emission of a

wide variety of pollutants in GD from 2006 to 2015 using a technology-based
methodology. The emission trends and their spatial variation were validated by ground-
based observations and satellite data. Anthropogenic emissions of most pollutants in
GD generally saw downward trends over the 2006-2015 decade, with $NH_3$ and VOC
emissions being the exceptions. In that decade, emissions of $SO_2$, $PM_{10}$, $PM_{2.5}$, and
$NO_X$ decreased by 48%, 22%, 16%, and 0.5%, respectively, despite significant growth
of economic and anthropogenic activity. The decoupling of anthropogenic emissions
from economic and energy consumption growth means that emission regulations and
control measures on power plants, industrial combustion, on-road mobile sources, and
dust sources enacted over the past decade have alleviated emissions. By contrast,
because of the absence of effective control measures, $NH_3$ emissions remained stable
while VOC emissions steadily increased by 33% during 2006-2015.

Because of their differences in economic and industrial structure and in their

implementation of control measures, PRD and NPRD showed different emission trends.
In PRD, $SO_2$ and $NO_X$ emissions exhibited a downward trend during 2006-2015, but in
NPRD, these emissions grew before 2010. Most of the increased emissions were from
power plants, industrial combustion, and non-road mobile sources, highly associated
with the shift of industries and power plants from PRD to NPRD and the lack of
stringent emission control measures in NPRD. The evolution of sources also showed
differences between PRD and NPRD. In PRD, emissions from industrial combustion
declined consistently during 2006-2015 owing to stringent control measures—e.g.
phasing out small boilers, installing flue-gas desulphurization (FDG) equipment, and
limiting the sulfur contents of fuel—while in NPRD, these emissions continued to show
an upward trend, even though some emissions from industrial combustion had been
reduced by control measures. Similar to the situation with industrial combustion,





emissions from industrial processes also declined in PRD but increased in NPRD during 2006-2015. The above-mentioned trends inevitably changed the relative contribution of different sources. Industrial combustion surpassed power plants as the largest contributor to $SO_2$ emissions in PRD and NPRD. For $NO_X$ emissions, on-road mobile sources were still the largest source in PRD, but in NPRD, the contribution of industrial combustion steadily increased and might replace power plants as the key contributor in the future. As to VOC emissions, industrial solvent use was the largest contributor in PRD, but in NPRD, the contribution of non-industrial solvent use increased and became the largest contributor.

The historical emission inventory developed in this study not only helps to understand the emission evolution in GD, but also can help to develop robust control measures for the co-control of $PM_{2.5}$ and ozone. In GD, future work should focus on power plants, industrial combustion, and non-road mobile sources to further reduce emissions of $SO_2$, $NO_X$, and particulate matter. This can be achieved by technical innovations consisting of ultra-low emissions in power plants, a series of actions regarding boiler management, control of sulfur content in fuels, flue-gas desulfurization in industries, and special pollution controls for non-metallic minerals industries. In addition, control measures on agricultural sources, the largest contributors of $NH_3$ emissions, should be highlighted. As revealed by Yin et al. (2018), the chemical region in PRD might transit to an ammonia-rich region with the decrease of $SO_2$ and $NO_X$ emissions. In this case, a larger reduction in $NH_3$ emissions would be required to further decrease ambient $PM_{2.5}$ levels in GD. This is feasible since $NH_3$ emissions in GD still have great potential for further reductions. In order to achieve co-control of $PM_{2.5}$ and ozone, future work should also focus on VOC emissions. In fact, the reduction of VOCs emissions is promising since stringent controls on solvent use was released in *Volatile organic compounds (VOCs) remediation and emission reduction work plan in Guangdong Province (2018-2020)* (GDEP, 2018). Apart from regulating solvent use, control measures for on-road mobile sources should be enhanced to cover the growth of emissions induced by the increase of vehicle population.

A long-term historical emission inventory could also help to reveal the dominant causes of air-quality change. In PRD, the annual averaged $PM_{2.5}$ concentrations showed a decrease in the 2006-2015 decade, from 58 $\mu g/m^3$ in 2007 to 34 $\mu g/m^3$ in 2017. By contrast, the 90th-percentile daily max 8-h average ozone showed a fluctuating increase, from 146 $\mu g/m^3$ in 2007 to 165 $\mu g/m^3$ in 2017. Our proposed a long-term historical inventory might be able to explain the change of $PM_{2.5}$ and ozone concentrations. As shown in Fig. 5, emissions of $SO_2$, $NO_X$, and $PM_{2.5}$ in PRD all steadily fell in this



decade. Particularly, $SO_2$ and $NO_X$ emissions, the two major precursors of $PM_{2.5}$
formations, decreased by 63% and 14%, respectively, during 2006-2015. This trend of
precursor emissions agreed with the declining trend of ambient $PM_{2.5}$ concentrations.
VOC emissions in PRD showed a rising trend, increasing by 35% during 2006-2015.
Ou et al. (2017) had revealed that most parts of PRD formed a VOC-limited region in
autumn and winter. This suggests that the growing VOC emissions and the decreasing
$NO_X$ emissions might contribute to the growth of ozone concentrations in PRD.
However, this does not mean that emission changes are the dominant cause. Using
numerical simulations and the long-term historical emission inventory developed in this
study, we can quantify the effectiveness of emission control measures and the impact
of meteorological change on air quality in PRD. Consequently, the dominate cause of
the increase of ambient ozone concentrations and the downward trend of $PM_{2.5}$
concentrations in PRD in the recent decade can be identified.

## Authorship Contribution Statement

Zheng J. Y., and Huang Z. J. provided writing ideas with Shao M. support. Huang
Z. J., Bian Y. H., and Ou J. M. carried them out. Huang Z. J., Zheng J. Y., and Ou J. M.
revised and polished the article. Bian Y. H., Zhong Z. M., Xiao X., Ye, X. and Wu Y. Q.
developed the decadal emission inventories and contributed to discussions of results.
Chen, L. F., Xu, Y. Q., Zhang, Z. W. and Yin, X. H. helped with verification of satellite
data. All authors have made substantial contributions to the work reported in the
manuscript.

## Competing interests

The authors declare that they have no conflict of interest.

## Acknowledgments

This study was supported by the National Key R&D Program of China (No.
2018YFC0213902), the NSFC National Distinguished Young Scholar Science Fund
(No. 41325020) and the National Key Technology Research and Development Program
of the Ministry of Science and Technology of China (No. 2014BAC21B02). The
authors thank the Institute of Remote Sensing and Digital Earth, Chinese Academy of



Sciences, for providing the satellite data.

## Supplementary information

Attached please find supplementary information associated with this article.





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





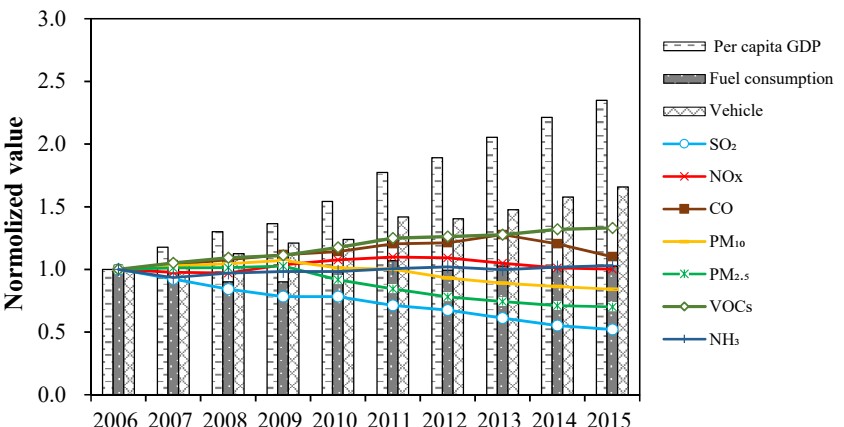

**Figure 1.** Trends in the air pollutant emissions, per capita GPD, fuel consumption and vehicle population in Guangdong Province from 2006 to 2015 (all of data are normalized to the year 2006).

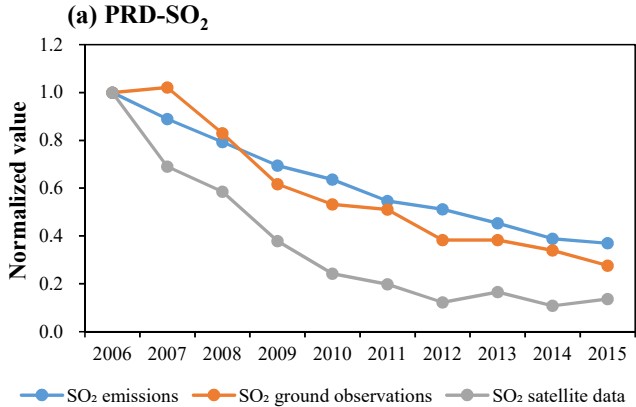





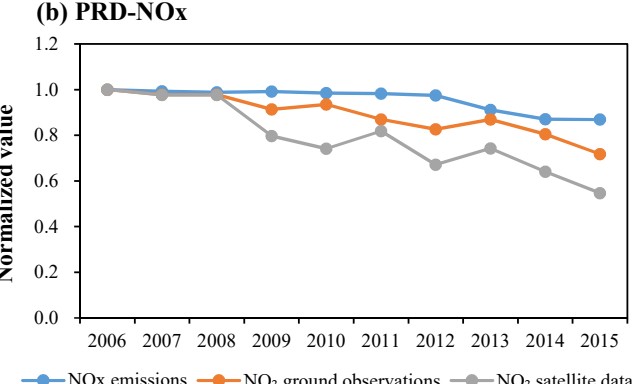

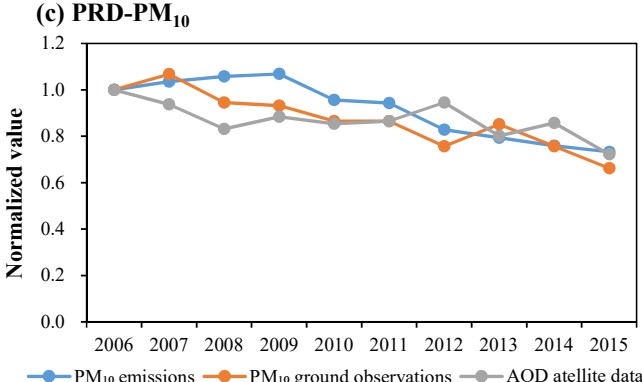

**Figure 2**. Comparison of emission trends with measurements in the PRD from 2006 to 2015. **(a)** $SO_2$ **(b)** $NO_X$ **(c)** $PM_{10}$.

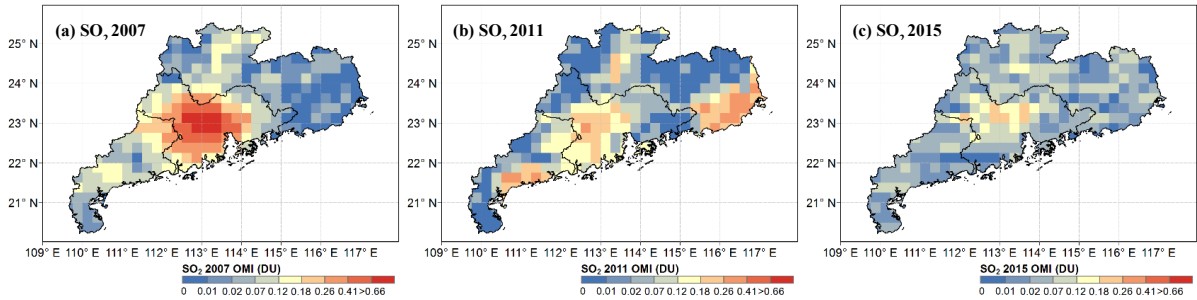





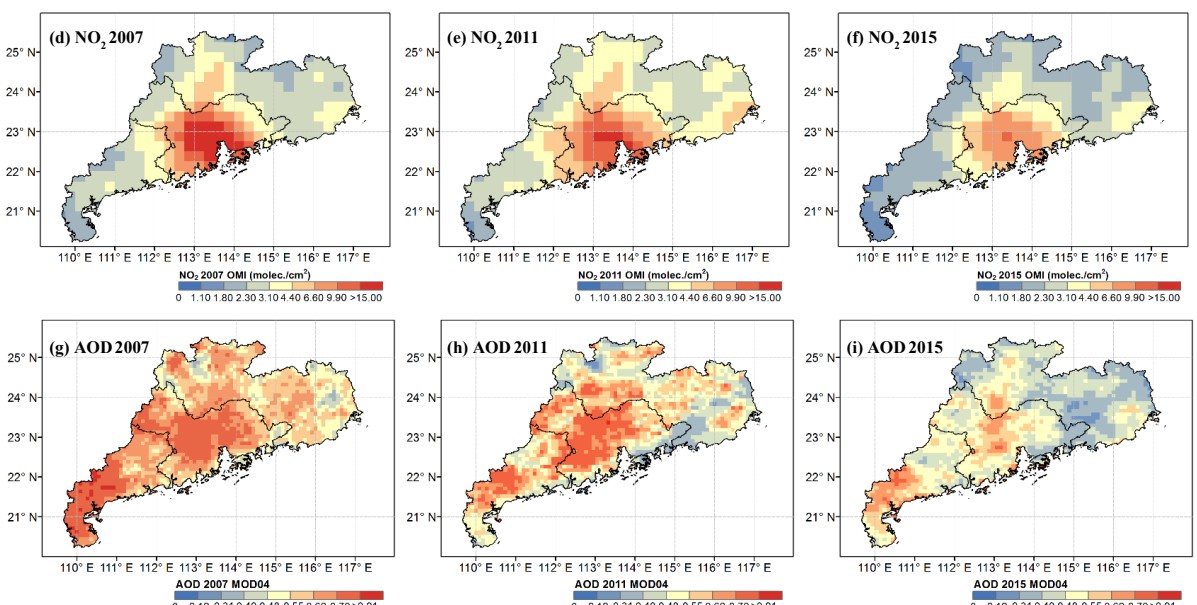

**Figure 3**. The spatial patterns of satellite observations over GD in 2007, 2011 and 2015 for **(a)-**

**(c)** SO₂, **(d)-(f)** NO₂, and **(g)-(i)** AOD. The legends represent a gradual increase in emissions from

right (blue) to left (red).



**Figure 4**. Source emission evolutions in Guangdong Province for **(a)** SO$_2$, **(b)** NO$_X$, **(c)** VOCs, **(d)** PM$_{10}$, **(e)** PM$_{2.5}$, **(f)** CO and **(g)** NH$_3$ from 2006 to 2015.





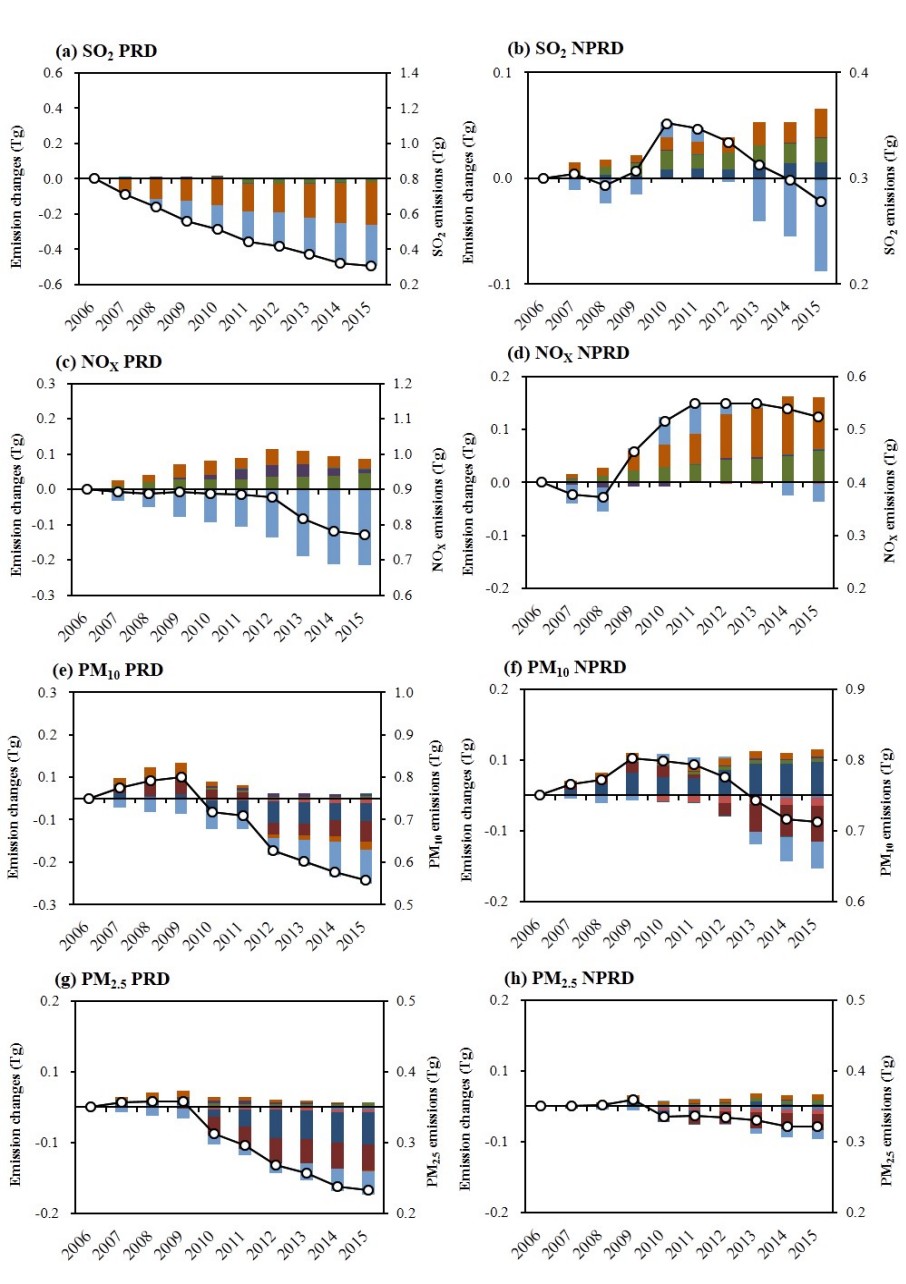



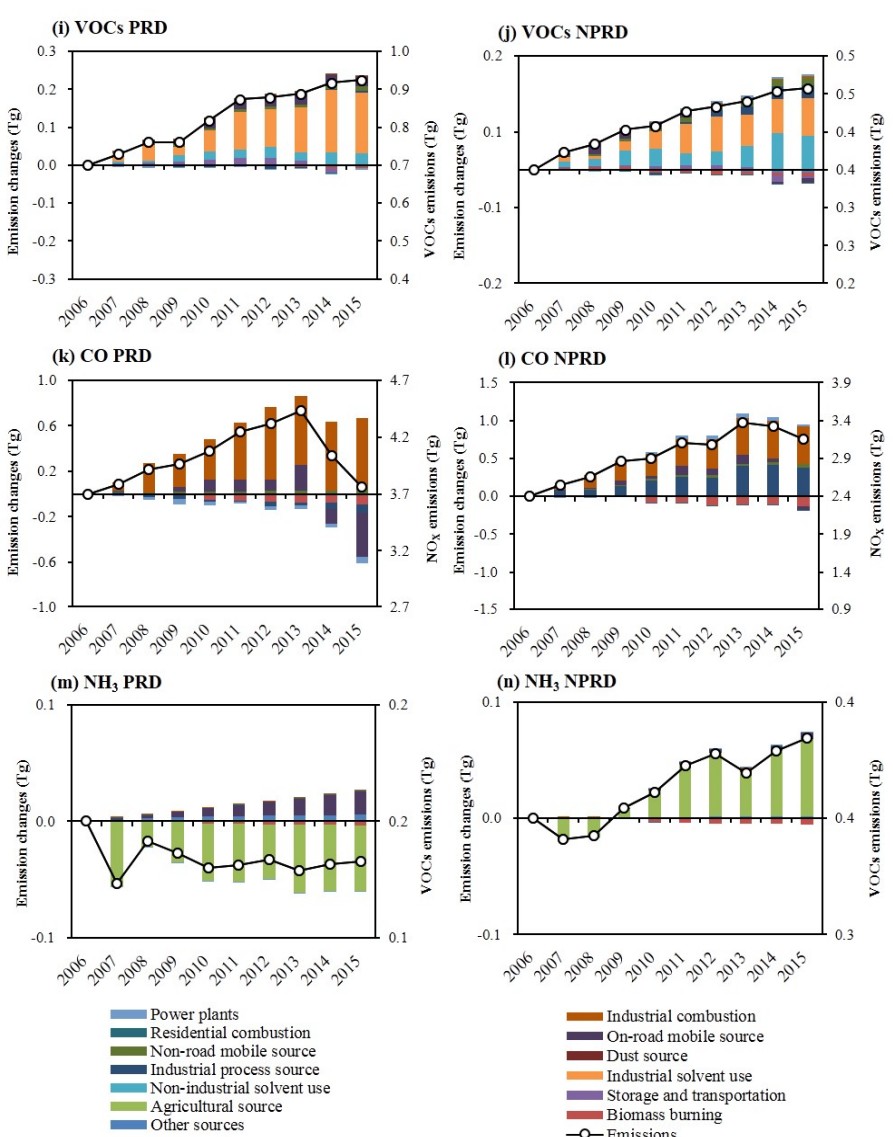

**Figure 5**. Emission evolutions by source in the PRD and NPRD for **(a)-(b)** $SO_2$, **(c)-(d)** $NO_X$, **(e)-(f)** $PM_{10}$, **(g)-(h)** $PM_{2.5}$, **(i)-(j)** VOCs, **(k)-(l)** CO and **(m)-(n)** $NH_3$ from 2006 to 2015. Source emissions in 2006 were subtracted from total emissions for each year to exhibit the additional emissions compared to 2006 (left axle). The total emissions by pollutant during 2006-2015 was also reflected in right axle.



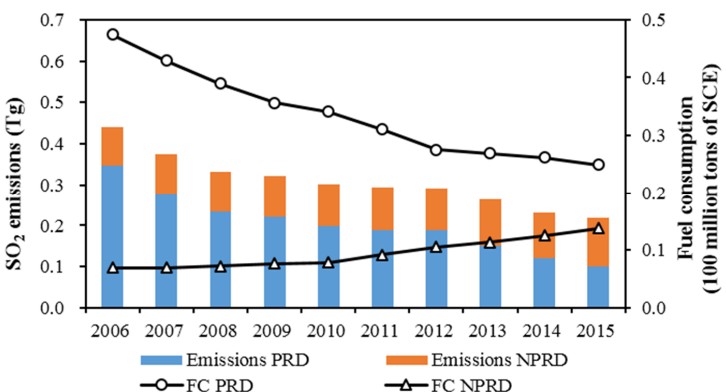

**Figure 6.** Trends of SO$_2$ emissions and fuel consumption from industrial combustion from 2006 to 2015. (SCE: standard coal equivalent; FC: fuel consumption).

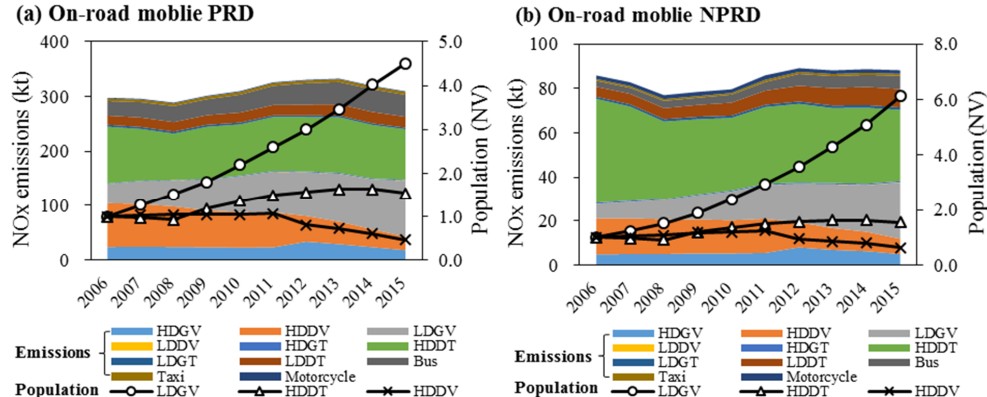

**Figure 7.** Trends of NO$_X$ emissions from on-road mobile source and its activity data from 2006 to 2015 in the **(a)** PRD and **(b)** NPRD.



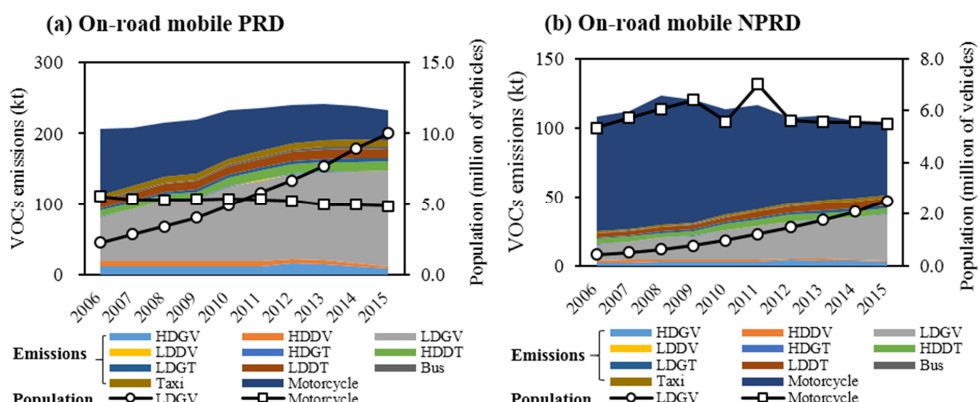

**Figure 8.** Trends of VOCs emissions from on-road mobile source and its activity data from 2006 to 2015 in the **(a)** PRD and **(b)** NPRD

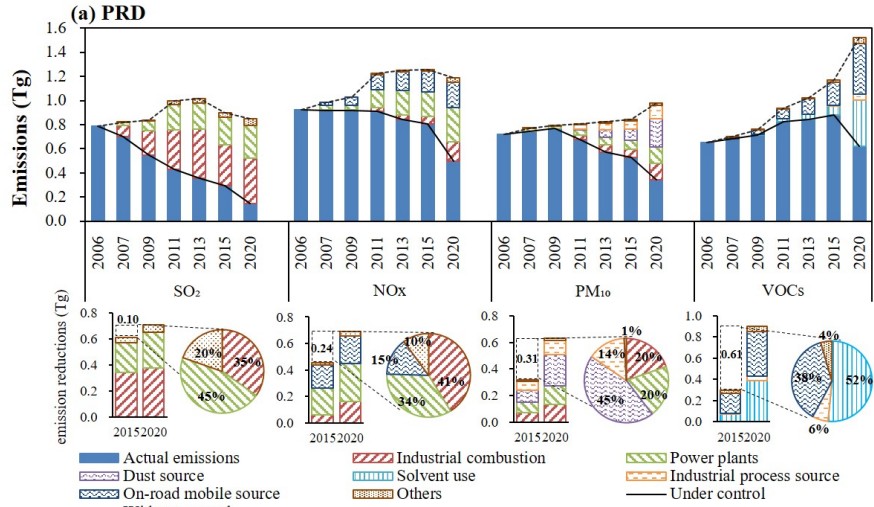



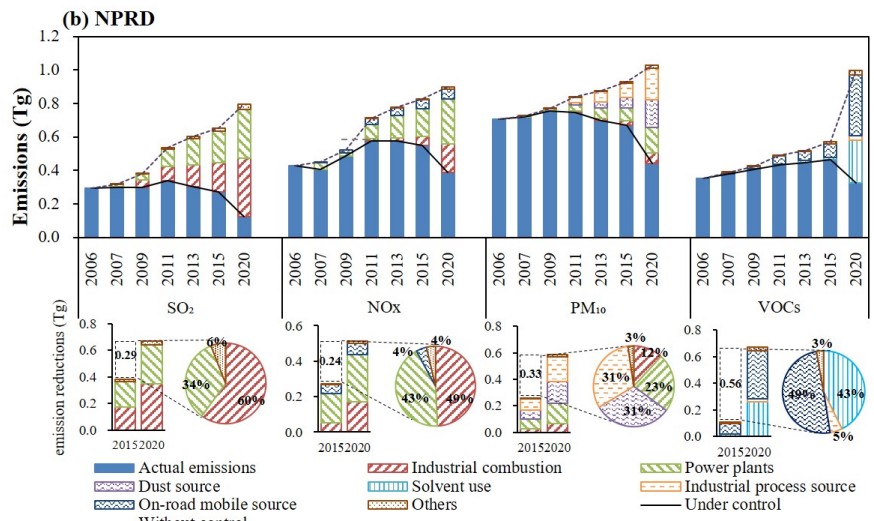

**Figure 9.** Control- and activity-driven emissions of SO$_2$, NO$_X$, PM$_{10}$ and VOCs in 2007, 2009, 2011, 2013, and 2015 in addition to emission predictions in 2020 for the **(a)** PRD and **(b)** NPRD. Solvent use here includes industrial solvent use and non-industrial solvent use. The solid black line and the solid blue bar denotes the actual emissions we estimated under control (i.e., the results under the interaction of control- and activity-driven emissions; if control-emission dominated, the emissions would drop, and vice versa), and the dotted black line denotes the hypothetical emissions without control (i.e., activity-driven emission; if no new control measures were adopted after 2006). The non-solid chromatic bars and pies illustrate the emission reductions for multiple sources (i.e., control-driven emission). The dotted box represents extra emission reductions in 2020 compared to 2015.