# Peer review of "Evolution of Anthropogenic Air Pollutant Emissions in"

_Atmospheric Chemistry and Physics, 2019_

## Referee Comment (RC1) · Anonymous Referee #1 · 17 Apr 2019

This work describes the long-term emission inventories for seven pollutants throughout Guangdong Province, one of most developed regions in China, which is highly significant. However, the paper still suffers for two detail problems. 1)The single and plural forms for source categories in the article are not very uniform. For example, the emission source –"dust sources" in line 147 is a plural form, but "dust source" is presented in Figure 4. There is an example taken here, which have also been the same problems in the paper. 2)The abscissa of figure 4(d) PM2.5 is diagonal. However, those for the other pollutants are vertical.

---

## Referee Comment (RC2) · Anonymous Referee #2 · 21 Apr 2019

**Review of "Evolution of Anthropogenic Air Pollutant Emissions in Guangdong Province, China, from 2006 to 2015"**

This paper calculated a multi-year emission inventory (from 2006 to 2015 and a future year 2020) using the method of activity data and the emission factors. And then the author discusses the emission trends, source contributions and pick out some examples to explain the background reasons. This is a good try to show the audience a data set and comparison. However, this paper is more like a government report instead of a top scientific research paper. There are no sound scientific findings in the manuscript. The results of the manuscript are almost a repeat or quite similar to previous publications:

(1). Zhong Z. (2018) Recent developments of anthropogenic air pollutant emission inventories in Guangdong province, China, Science of Total Environment.

(2). Yin, X. H., Huang, Z. J., Zheng, Y., Yuan, Z. B., Zhu, W. B., Huang, X. B., and Chen, D.H.: Source contributions to PM2.5 in Guangdong province, China by numerical modeling: Results and implications, Atmospheric Research.

Zhong's paper is comparing with 2006, 2010 and 2012 GuangDong emission inventory while the current manuscript is analyzing the data from 2006 to 2015. The source categories of these two papers are more or less the same although the author claimed in the manuscript that the source categories follows another publication: Pan et al. 2015 (line 144). The unabated emission factors of current manuscript are more or less the same with that in Zhong's paper. Moreover, there is inconsistency in factor values or total emission amount for a corresponding source/pollutants using the same model such as IVE model for mobile source. In terms of the emission validation using observation and satellite data, it is a good way to validate the bottom-up emission inventory utilizing this kind of data. However, the sparsity of the satellite needs to be considered. The carelessness comparison in this manuscript is very simple, not much extra information. Finally, although the manuscript is evaluating the control measures, The main results (figure 9) is more like a repeat of section 2.3 Evolution of source emissions in Guangdong Province for the

year 2006 to 2015. To sum up, I recommend a direct rejection because no meaningful results or new scientific findings can be found through the manuscript.

The following is the detailed comments for the manuscript.

1. **Abstract section:** Line 39-43: "The declines of SO2, NOx, PM2.5, and PM10 emissions are mainly attributed to the control-driven emission reductions in the Pearl River Delta (PRD) region, especially from power plants, industrial combustion, on-road mobile sources, and fugitive dust, and partly to the shift of industries from the PRD to the non-PRD (NPRD) region in GD." It does not show any new findings here, especially the idea of "shift of industries from the PRD to the non-PRD (NPRD) region in GD" comes from a published paper on Atmospheric Research (AR) titled: "Source contributions to PM2.5 in Guangdong province, China by numerical modeling: Results and implications" by X. Yin (2017)

   Line46-48: "this might be one of the reasons that led to the slight upward trends of ozone concentrations in GD." is this finding a reliable result or just a guess? It is not appropriate to put an ambiguous answer in the central part of the abstract.

2. **Introduction section**:
   **Overall review:** For a publication in a top journal such as ACP, the research motivation of the current manuscript is not strong enough. The literature review is too weak. Too many self-cited references were present.

   Line 84-91: I do not see the long term emission inventory comparison could resolve the controversial issue of whether the emission or the meteorology plays the main role of pollution. Please explain how the long term emission inventory could help differentiate the causes of the emission or the meteorology?

   Line 96-112 introduces the history of emission inventory development in PRD and GD. Too many self-cited papers are included. Due to differences in source categories or data

sources for different geometry, using a unified method to calculate a multi-year emission inventory is fine, but what is new here? Is the technique original or are the data sources new? What are the advantages/Strength of the method/data in this manuscript comparing with the previous publications? Only calculation of a multi-year EI seems too weak to be published in a top journal ACP.

3. **Methodology section.**

**General review:** No new method or new data sources were found in this manuscript comparing with the previous Emission Inventory publications in the PRD region. The calculation of marine emission is outdated, and its uncertainty is considerable. The validation method of the emission totals is too simple to be believed.

Table S5: is the table for PM10? Please be specific.

Line 212 – 215: The dealing method of marine may under-estimate the marine emissions which are quite important for Guang Dong coastal cities.

Line 222- 223: Please specify how typical are the annual average vehicle kilometers? How many cities and what types of roads in Guang Dong were involved in the field survey?

Line: 225-227: The vehicle ratios of fuel use used in this manuscript is too outdated to be used. Moreover, the vehicle ratios should be differentiated for PRD and NPRD. The pattern of the vehicle population in GD changed quite a lot from 2009 to 2015.

4. **Results section**

**General review:** It is just the description of the changes and the sectoral contribution of the multi-year emission inventory. Relevant policy measures were used to explain the sharp drops or increases. It is more like a government report instead of a research paper published in a top journal.

Line 343: I did not see "emission trends in PRD and NPRD are shown in Fig. 2", Fig 2 only show the trend of PRD instead of NPRD. Please specify.

Line 346-348: PM2.5 was not shown in figure 2. Please specify.

Line 349-365: Spatial maps of differences between different years (Figure 3) for the validation are better to see the changes. In term of calculating the emission changes, did you count all the grids or just the typical points in PRD and NPRD region? Please specify.

Line 395: figure 5a is showing the result of PRD, not GD. Please specify.

Line 437: "On-road mobile was also a major contributor to NOX emissions in GD (Fig. 5b)." Fig. 5b is showing results of SO2 for NPRD. Please double check.

Line 673: it should be 2.4.4

---

## Author Comment (AC1) · 21 May 2019

Thanks for the valuable comments and positive feedback. We had corrected all the typo errors in the manuscript. Below are the point-to-point responses to two specific comments:

The single and plural forms for source categories in the article are not very uniform. For example, the emission source –"dust sources" in line 147 is a plural form, but "dust source" is presented in Figure 4:

Response: Thanks for the comments. We had unified the source categories through

out the manuscript, which are: power plants, industrial combustion, residential combustion, on-road mobile source, non-road mobile source, dust source, industrial process source, industrial solvent use, non-industrial solvent use, storage and transportation, agricultural source, biomass burning, and other sources.

The abscissa of figure 4(d) PM2.5Âň is diagonal. However, those for the other pollutants are vertical.

Response: Thanks for pointing out the typo error. We had corrected the error accordingly in the revised manuscript (we attached the figure 4 here).

[Figure]

**Fig. 1.**

---

## Short Comment (SC1) · 27 May 2019

This paper shows the long-term evolution (2006~2015) of anthropogenic emissions in Guangdong Province of China, to reveal the causes of source-contribution changes during this periods. The conclusions obtained in this paper are of great significance for the deep understanding of regional pollution emission characteristics, and have important guiding for co-control strategies on regional atmospheric pollution. The datasets produced by the article are conducive to the in-depth study of scientific issues in related fields. Thereforeï¡Ňthis paper is strongly recommended for publication.

[Figure]

2019.

---

## Author Comment (AC2) · 28 May 2019

Response to the Reviewer 2

General commentsïijŽ This paper calculated a multi-year emission inventory (from 2006 to 2015 and a future year 2020) using the method of activity data and the emission factors. And then the author discusses the emission trends, source contributions and pick out some examples to explain the background reasons. This is a good try to show the audience a data set and comparison. However, this paper is more like a government report instead of a top scientific research paper. There are no sound scientific findings in the manuscript. The results of the manuscript are almost a repeat

or quite similar to previous publications: (1). Zhong Z. (2018) Recent developments of anthropogenic air pollutant emission inventories in Guangdong province, China, Science of Total Environment. (2). Yin, X. H., Huang, Z. J., Zheng, Y., Yuan, Z. B., Zhu, W. B., Huang, X. B., and Chen, D.H.: Source contributions to PM2.5 in Guangdong province, China by numerical modeling: Results and implications, Atmospheric Research. Zhong's paper is comparing with 2006, 2010 and 2012 GuangDong emission inventory while the current manuscript is analyzing the data from 2006 to 2015. The source categories of these two papers are more or less the same, although the author claimed in the manuscript that the source categories follows another publication: Pan et al. 2015 (line 144). The unabated emission factors of current manuscript are more or less the same with that in Zhong's paper. Moreover, there is inconsistency in factor values or total emission amount for a corresponding source/pollutants using the same model such as IVE model for mobile source. In terms of the emission validation using observation and satellite data, it is a good way to validate the bottom-up emission inventory utilizing this kind of data. However, the sparsity of the satellite needs to be considered. The carelessness comparison in this manuscript is very simple, not much extra information. Finally, although the manuscript is evaluating the control measures, The main results (figure 9) is more like a repeat of section 2.3 Evolution of source emissions in Guangdong Province for the 2 year 2006 to 2015. To sum up, I recommend a direct rejection because no meaningful results or new scientific findings can be found through the manuscript.

Response: We would like to thank the reviewer's valuable time and effort in reviewing the manuscript.

1) "This paper is more like a government report instead of a top scientific research paper. There are no sound scientific findings in the manuscript". Such an impression might be raised when we introduced the emissions and their trends in Section 2.3 and 2.4, which is somehow inevitable in a data-intense paper. However, it does not necessarily mean that it cannot be a top scientific research paper. The importance

of long-term historical emission inventories in air pollution research have been widely recognized by the academic community (Li et al., 2017; Zhang et al., 2018; Hosely et al. 2018; Jin et al. 2017; Li et al. 2017; Lei et al. 2011; Ohara et al. 2017). Here, we advance the understanding of long-term air pollutant emissions in China. The study area, Guangdong province, is one of the three largest city clusters in China. It plays an important role in China's air pollution research frontiers and mitigation efforts. Surprisingly, no attempts have been made to estimate its historical emissions. Some studies have estimated the emissions of the Pearl River Delta (PRD) region, but the emissions of PRD cannot fairly represent Guangdong as a whole. To our best knowledge, this work provides the first long-term record of anthropogenic air pollutant emissions of Guangdong. It provides robust data to assess the effectiveness of control measures and unveil the primary cause of air quality change and fills an important gap in China's air pollution research. Thus, it is potentially highly cited by future work in South China. Moreover, this work observes the phenomenon of 'emission leakages' from developed cities to less developed ones. Emissions in Non-PRD (the other less-developed cities in Guangdong) are becoming increasingly important with some emerging sources such as industrial processes. A new clue or an alarm is putting forward, in sharp contrast to the current mitigation efforts focusing on developed regions.

2) "The results of the manuscript are almost a repeat or quite similar to previous publications (Zhong et al., 2018; Yin et al., 2017)". We appreciate reviewer's efforts in literature review, but we have to point out that our work is quite different from them regarding methodologies, study areas and findings. Zhong et al. (2018) summarized recent updates of regional emission inventories in Guangdong in terms of the emission source supplement, spatiotemporal distribution refinement and estimation method. Yin et al (2017) used an atmospheric model to quantify the source contributions to ambient PM2.5 other than PM2.5 emissions while the 2012 Guangdong emission inventory was used as model input. In comparison, our study applied a dynamic technology-based methodology to develop a long-term (2006-2015) emission inventory in Guangdong and analyzed its emission trends, spatial variations, source-contribution variations, and

emission reduction potentials. The study area is another difference. While Zhong et al. (2018) compared the PRD emission inventories using different estimation methods as an example (not the Guangdong emission inventories as claimed by the reviewer: Zhong's paper is comparing with 2006, 2010 and 2012 Guangdong emission inventory), this study estimated long-term emissions for Guangdong using self-consistent methodologies. Also, this study compared the emission evolution between the PRD and Non-PRD regions, and found that the emission evolution in the Non-PRD is quite different from their counterparts in PRD, which are not revealed by two previous studies. Therefore, it is arbitrary and hasty to conclude that our work is a 'repeat' of Zhong et al (2018) and Yin et al (2017).

3) "The source categories of these two papers are more or less the same although the author claimed in the manuscript that the source categories follows another publication: Pan et al. 2015 (line 144). The unabated emission factors of current manuscript are more or less the same with that in Zhong's paper. Moreover, there is inconsistency in factor values or total emission amount for a corresponding source/pollutants using the same model such as IVE model for mobile source". In fact, Zhong et al. (2018) updated the source categories based on Pan et al. (2015). Therefore, it is not surprising that the source categories of our study and Zhong et al. are similar. However, I am afraid that the classification similarity does not mean these papers are repetitive. Instead, a similar source category is beneficial to the evaluation of our study with previous studies. In response to the comment, we added a clarification in the method section 1.1. "We estimated emissions of 7 pollutants (SO2, NOX, PM10, PM2.5, VOCs, CO, and NH3) from 13 major categories and 70 sub-categories based on Zhong et al.(2018), Pan et al. (2015) and the guidelines for the development of an air-pollutant emission inventory for Chinese cities (MEPC, 2017), in order to make this study could be comparable with previous emission inventories.". Regarding the vehicle emission factors, the inconsistency arises because the base years of unabated emission factors in this study is 2007, rather than 2006 in Zhong et al. (2018).

4) "In terms of the emission validation using observation and satellite data, it is a good way to validate the bottom-up emission inventory utilizing this kind of data. However, the sparsity of the satellite needs to be considered. The carelessness comparison in this manuscript is very simple, not much extra information." We admit that the evaluation is simple, but is effective. Evaluation of a long-term emission inventory should focus on its emission trend, rather than the amount or high-resolution spatial pattern. Therefore, the sparsity of the satellite data and ground-level observations is not a concern in the evaluation. In fact, similar evaluation was also adopted by Zhang et al.(2018), which compared the trends in PM2.5 precursor emissions in China with satellite and ground-based PM2.5 concentrations, indicating the evaluation is acceptable. In response to the questioning, we revised the method section 1.2.

5) "the main results (figure 9) is more like a repeat of section 2.3 Evolution of source emissions in Guangdong Province". We're afraid the conclusion is improper. Section 2.3 compares how the emissions in PRD and non-PRD are evolving in different ways. Figure 9 studies how might the emission changes in the absence of control measures and reduction potentials if actions have been taken, under the future projection of activity level. Combining section 2.3 and Figure 9 could provide a better understanding of emission trends. For example, we observed a slight change of mobile VOC emission in Section 2.3. In fact, control measures were effective in reducing mobile VOC emission in the past decade. Nevertheless, the emission reduction was compensated by the growth of the vehicle population as revealed by Figure 9. Emission potentials are also quantified in Figure 9, which is not included in Section 2.3.

Detailed comments:

1. Abstract section: Line 39-43: "The declines of SO2, NOx, PM2.5, and PM10 emissions are mainly attributed to the control-driven emission reductions in the Pearl River Delta (PRD) region, especially from power plants, industrial combustion, on-road mobile sources, and fugitive dust, and partly to the shift of industries from the PRD to the non-PRD (NPRD) region in GD." It does not show any new findings here, especially

the idea of "shift of industries from the PRD to the non-PRD (NPRD) region in GD" comes from a published paper on Atmospheric Research (AR) titled: "Source contributions to PM2.5 in Guangdong province, China by numerical modeling: Results and implications" by X. Yin (2017).

Response: We are afraid that this is a misunderstanding. In Yin et al. (2018), the atmospheric modeling results showed that emissions in NPRD could contribute to the PM2.5 pollution in PRD. Therefore, they suggested that relocating industries from PRD to NPRD region cannot help ease the pollution in PRD. In short, Yin et al. (2018) discussed one possibility of emission control and proved that it might not work. In our study, emission leakages from PRD to NPRD are observed through a multi-year record of emissionsïijŇwhich is not revealed by Yin et al. (2018).

Line46-48: "this might be one of the reasons that led to the slight upward trends of ozone concentrations in GD." is this finding a reliable result or just a guess? It is not appropriate to put an ambiguous answer in the central part of the abstract.

Response: Thanks for the comment. Ou et al. (2016) revealed that O3 formation in most parts of PRD was VOC-limited in autumn and winter. This indicates that the growing VOC emissions and the decreasing NOX emissions, which was observed in this study, might contribute to the growth of ozone concentrations in PRD. However, this does not mean that the emission change is the dominant cause of ozone growth. Further studies using numerical simulations are required to verify the guess. In fact, our preliminary simulations confirm that the growing VOC emissions and the decreasing NOX emissions account for about 90% of the ozone growth in PRD in the past decade. Since this result has not yet been published, we decide to removed "this might be one of the reasons that led to the slight upward trends of ozone concentrations in GD" in the revised manuscript.

Reference: Ou, J. Y., Yuan, Z. B., Zheng, J. Y., Huang, Z. J., Shao, M., Li, Z. K., Huang, X. B., Guo, H., and Louie, P. K. K.: Ambient Ozone Control in a Photochemically Active

Region: Short-Term Despiking or Long-Term Attainment? Environmental Science & Technology, 50(11):5720-5728, doi: 10.1021/acs.est.6b00345, 2016.

2. Introduction section: Overall review: For a publication in a top journal such as ACP, the research motivation of the current manuscript is not strong enough. The literature review is too weak. Too many self-cited references were present.

Response: Thanks for the comment. The study area, Guangdong province, is one of the three largest city clusters in China. It plays an important role in China's air pollution research frontiers and mitigation efforts. Subsequently, Guangdong has experienced significant air quality improvement, particularly the Pearl River Delta region, which is the first region to meet China's national 35 $\mu$g/m3 PM2.5 standard for four consecutive years. Understanding the emission evolution and assessing the effectiveness of control measures call for a systematic review of historical emissions in Guangdong. Surprisingly, no attempts have been made to estimate Guangdong's historical emissions. In this study, we developed a long-term historical emission inventory in Guangdong. To our best knowledge, this work provides the first long-term record of anthropogenic air pollutant emissions of Guangdong. Also, our study observes the phenomenon of 'emission leakages' from the developed cities to less developed ones, and the different emission evolution pattern between the Non-PRD and PRD regions. Combining with numerical model simulations, the long-term emission data developed in this study can identify the cause of air quality change in Guangdong. Similar works had been conducted in Beijing and published in ACP (Cheng et al. 2018). Our team also used the long-term emission data and numerical simulation to reveal the dominant cause of ozone changes in PRD. In response to the comment, we will revise the Introduction to highlight the scientific significance of this study. Regarding the self-cited references, we think this is inevitable since most emission inventories in PRD were developed by our team.

Referenceïjž Cheng, J., Su, J., Cui, T., Li, X., Dong, X., Sun, F., . . . He, K. (2018). Dominant role of emission reduction in PM2.5; air quality improvement in Beijing during 2013-2017: a model-based decomposition analysis. Atmospheric Chemistry and Physics Discussions, 1–31. https://doi.org/10.5194/acp-2018-1145

Line 84-91: I do not see the long-term emission inventory comparison could resolve the controversial issue of whether the emission or the meteorology plays the main role of pollution. Please explain how the long-term emission inventory could help differentiate the causes of the emission or the meteorology?

Response: Thanks for the comment. A long-term emission inventory alone could not identify the cause of air quality change. However, combining with atmospheric chemical transport models (CTMs), the impact of emission change and meteorological variation on air quality can be quantified. For instance, by combining a long-term emission and the WRF-CMAQ (Weather Research and Forecasting Model and Community Multiscale Air Quality) model, Cheng et al (2018) found that the rapid decrease in PM2.5 concentrations in Beijing during 2013–2017 was dominated by local and regional emission reductions, rather than meteorology variation. In response to the comment, we will revise the expression for clarification.

Reference: Cheng, J., Su, J., Cui, T., Li, X., Dong, X., Sun, F., . . . He, K. (2018). Dominant role of emission reduction in PM2.5; air quality improvement in Beijing during 2013-2017: a model-based decomposition analysis. Atmospheric Chemistry and Physics Discussions, 1–31. https://doi.org/10.5194/acp-2018-1145

Line 96-112 introduces the history of emission inventory development in PRD and GD. Too many self-cited papers are included. Due to differences in source categories or data sources for different geometry, using a unified method to calculate a multi-year emission inventory is fine, but what is new here? Is the technique original or are the data sources new? What are the advantages/Strength of the method/data in this manuscript comparing with the previous publications? Only calculation of a multi-year EI seems too weak to be published in a top journal ACP.

Response: Thanks for the comment. Following previous long-term emission papers

(Lu et al. 2012, Zheng, et al. 2018), we applied a dynamic technology-based method-ology that considers economic development, technological penetration, and emission controls to estimate the long-term emission inventory in Guangdong in this study. It is true that the methods and data sources are not new. However, these methods satisfy the requirement of developing a reliable long-term emission inventory. Although new data sources, such as AIS data and satellite data, can promote emission inventories, obtaining new data span a long-term period is challenging. Besides, for most long-term emission inventory papers, their scientific value lies in revealing the long-term emission evolution pattern, assessing the effectiveness of past control measures and providing guidelines for future control measure development. This study provides the first long-term record of anthropogenic air pollutant emissions of Guangdong, the three largest city clusters in China, and advances our understating of air pollutant emissions and control measures in Guangdong. For instance, this study observes the phenomenon of 'emission leakages' from the PRD region to the Non-PRD region, which received less attention of air quality control compared with the PRD, indicating that emissions in Non-PRD (the other less-developed cities in Guangdong) are becoming increasingly important in Guangdong. Also, this study found that past control measures for vehicle source and solvent use source are not stringent enough because emission reduction driven by these measures cannot cover the emission growth due to the increasing vehicle population and solvent use. Combining with atmospheric chemical transport models (CTMs), the long-term emission data developed in this study could help iden-tify the dominant cause of air quality change in Guangdong.

References: 1. Lu, Q., Zheng, J. Y., Ye, S. Q., Shen, X. L., Yuan, Z. B., and Yin, S. S.: Emission trends and source characteristics of SO2, NOX, PM10 and VOCs in the Pearl River Delta region from 2000 to 2009, Atmospheric Environment, 76, 11-20, doi: 10.1016/j.atmosenv.2012.10.062, 2013. 2. Zheng, B. , Dan, T. , Meng, L. , Fei, L. , Chaopeng, H. , & Guannan, G. , et al. (2018). Trends in china's anthropogenic emissions since 2010 as the consequence of clean air actions. Atmospheric Chemistry and Physics Discussions, 1-27.

3. Methodology section. General review: No new method or new data sources were found in this manuscript comparing with the previous Emission Inventory publications in the PRD region. The calculation of marine emission is outdated, and its uncertainty is considerable. The validation method of the emission totals is too simple to be believed.

Response: Thanks for the comment. Unlike the single-year emission inventory, the estimation of a long-term emission inventory is generally more complicated since it requires the data sources, estimation method and source category for all years are consistent. Although new estimation methods and new data sources can promote emission inventories, obtaining new data span a long-term period is challenging. For example, we understand that the bottom-up estimation method based on AIS data can generate reliable ship emission inventories. However, obtaining 10-year AIS data is considerable work. Moreover, the spatial coverage of AIS data in China before 2012 is limited. Instead, long-term ship fuel consumption data or cargo volumes can be easily obtained. Thus, the top-down method based on fuel consumption or cargo volumes is more feasible than the bottom-up method based on AIS data in the long-term emission estimation, although the uncertainty is larger. Moreover, long-term emission inventories generally focus on emission trend, sectoral evolution, and emission projection, rather than the high-resolution spatial distribution. Therefore, most studies (Streets et al. 2006, Zhang et al, 2007, Lu et al. 2012, Zheng, et al. 2018) still generally applied the top-down method to develop long-term emission inventories. In this study, we applied a dynamic technology-based methodology that considers economic development, technological penetration, and emission controls to estimate the long-term emission inventory in Guangdong. Also, we used interpolation method and fuel consumption estimation method to estimate some missing data, make sure all activity data during 2006-2015 are comparable.

References: 1. Streets, D. G., Zhang, Q., Wang, L. T., He, K. B., Hao, J. M., Wu, Y., Tang, Y. H., and Carmichael, G. R.: Revisiting China's CO emissions after the Transport and Chemical Evolution over the Pacific (TRACE-P) mission: Synthesis of inventories, atmospheric modeling, and observations, Journal of Geophysical Research Atmospheres. 111, doi: 10.1029/2006JD007118, 2006. 2. Zhang, Q., Streets, D. G., He, K. B., Wang, Y. X., Richter, A., Burrows, J. P., Uno, I., Jang, C. J., Chen, D., and Yao, Z. L., NOX emission trends for China, 1995–2004: The view from the ground and the view from space, Journal of Geophysical Research: Atmospheres, 112, doi: 10.1029/2007JD008684, 2007. 3. Lu, Q., Zheng, J. Y., Ye, S. Q., Shen, X. L., Yuan, Z. B., and Yin, S. S.: Emission trends and source characteristics of SO2, NOX, PM10 and VOCs in the Pearl River Delta region from 2000 to 2009, Atmospheric Environment, 76, 11-20, doi: 10.1016/j.atmosenv.2012.10.062, 2013. 4. Zheng, B. , Dan, T. , Meng, L. , Fei, L. , Chaopeng, H. , & Guannan, G. , et al. (2018). Trends in china's anthropogenic emissions since 2010 as the consequence of clean air actions. Atmospheric Chemistry and Physics Discussions, 1-27.

Table S5: is the table for PM10? Please be specific.

Response: Yes. We will revise the typo error accordingly.

Line 212 – 215: The dealing method of marine may under-estimate marine emissions, which are quite important for Guang Dong coastal cities.

Response: Thanks for the comment. It is true that the top-down method based on activity data from statistic yearbook might under-estimate the marine emissions, compared with those estimated by the bottom-up method based on AIS data. However, the bottom-up method is less feasible in the estimation of long-term emission inventory due to the lack of long-term AIS data. In response to the comment, we will discuss the underestimation of marine emissions in this study.

Line 222- 223: Please specify how typical are the annual average vehicle kilometers? How many cities and what types of roads in GuangDong were involved in the field survey?

Response: Thanks for the comment. We obtained eight cities and 111 roads by types

in Guangdong province in the field survey. The road type includes arterial road, secondary arterial road, and branch, covering most road types in Guangdong.

Line: 225-227: The vehicle ratios of fuel use used in this manuscript is too outdated to be used. Moreover, the vehicle ratios should be differentiated for PRD and NPRD. The pattern of the vehicle population in GD changed quite a lot from 2009 to 2015.

Response: Thanks for the comment. In fact, we considered the annual change of vehicle ratios of fuel use based on the field survey of vehicle ratios that covered eight cities in Guangdong for the years of 2010, 2012, 2014, and 2015 and published data from Che et al. (2009) for the year of 2006. For other years, we estimated the vehicle ratios using an interpolation method. We agree that the vehicle ratios between PRD and NPRD are different. However, due to the limited survey data, we used the same vehicle ratio for PRD and NPRD regions, which could bring uncertainty in emission estimation. In response to the comment, we will clarify the data and discuss the limitation in the method section. Reference: Che, W. W., Zheng, J. Y., and Zhong, L. J.: Vehicle Exhaust Emission Characteristics and Contributions in the Pearl River Delta Region, Research of Environmental Sciences, 22, 456-461, 2009.

4. Results section General review: It is just the description of the changes and the sectoral contribution of the multi-year emission inventory. Relevant policy measures were used to explain the sharp drops or increases. It is more like a government report instead of a research paper published in a top journal.

Response: Thanks for the comment. Such an impression might be raised when we introduced the emissions and their trends in Section 2.3 and 2.4, which is somehow inevitable in a data-intense paper. However, it does not necessarily mean that it cannot be a top scientific research paper. The importance of long-term historical emission inventories in air pollution research has been widely recognized by the academic community (Li et al., 2017; Zhang et al., 2018; Hosely et al. 2018; Jin et al. 2017; Li et al. 2017; Lei et al. 2011; Ohara et al. 2017). Here, we advance the understanding of longterm air pollutant emissions in China. The study area, Guangdong province, is one of the three largest city clusters in China. It plays an important role in China's air pollution research frontiers and mitigation efforts. Surprisingly, no attempts have been made to estimate its historical emissions. Some studies have estimated the emissions of Pearl River Delta (PRD), but the emissions of PRD cannot fairly represent Guangdong as a whole. To our best knowledge, this work provides the first long-term record of anthropogenic air pollutant emissions of Guangdong. It provides robust data to assess the effectiveness of control measures and unveil the primary cause of air quality change and fills an important gap in China's air pollution research. Thus, it is potentially highly cited by future work in South China. Moreover, this work observes the phenomenon of 'emission leakages' from developed cities to less developed ones. Emissions in Non-PRD (the other less-developed cities in Guangdong) are becoming increasingly important with some emerging sources such as industrial processes. A new clue or an alarm is putting forward, in sharp contrast to the current mitigation efforts focusing on developed regions. In response to the comment, we revised the introduction and summary section to highlight scientific significance.

References: 1. Li, M.; Liu, H.; Geng, G.; Hong, C.; Liu, F.; Song, Y.; Tong, D.; Zheng, B.; Cui, H.; Man, H.; et al. Anthropogenic Emission Inventories in China: A Review. Natl. Sci. Rev. 2017, 4 (6), 834–866. 2. Hoesly, R. M.; Smith, S. J.; Feng, L.; Klimont, Z.; Janssens-Maenhout, G.; Pitkanen, T.; Seibert, J. J.; Vu, L.; Andres, R. J.; Bolt, R. M.; et al. Historical (1750-2014) Anthropogenic Emissions of Reactive Gases and Aerosols from the Community Emissions Data System (CEDS). Geosci. Model Dev. 2018, 11 (1), 369–408. 3. Jin, Q.; Fang, X.; Wen, B.; Shan, A. Spatio-Temporal Variations of PM2.5 Emission in China from 2005 to 2014. Chemosphere 2017, 183, 429–436. 4. Li, J.; Li, Y.; Bo, Y.; Xie, S. High-Resolution Historical Emission Inventories of Crop Residue Burning in Fields in China for the Period 1990-2013. Atmos. Environ. 2016, 138, 152–161. 5. Lei, Y.; Zhang, Q.; He, K. B.; Streets, D. G. Primary Anthropogenic Aerosol Emission Trends for China, 1990-2005. Atmos. Chem. Phys. 2011, 11 (3), 931–954. 6. Ohara, T.; Akimoto, H.; Kurokawa, J.; Horii, N.; Yamaji, K.; Yan, X.;

Hayasaka, T. An Asian Emission Inventory of Anthropogenic Emission Sources for the Period 1980–2020. Atmos. Chem. Phys. 2007, 7, 4419–4444. 7. Zheng, B. , Dan, T. , Meng, L. , Fei, L. , Chaopeng, H. , & Guannan, G. , et al. (2018). Trends in china's anthropogenic emissions since 2010 as the consequence of clean air actions. Atmospheric Chemistry and Physics Discussions, 1-27.

Line 343: I did not see "emission trends in PRD and NPRD are shown in Fig. 2", Fig 2 only show the trend of PRD instead of NPRD. Please specify.

Response: Thanks for pointing out this. The wrong expression for NPRD was removed in the manuscript.

Line 346-348: PM2.5 was not shown in figure 2. Please specify.

Response: Thanks for pointing out this. We made a typo error and will change "PM2.5" to "PM10" in the revised manuscript.

Line 349-365: Spatial maps of differences between different years (Figure 3) for the validation are better to see the changes. In term of calculating the emission changes, did you count all the grids or just the typical points in PRD and NPRD region? Please specify.

Response: Accepted, we will replot the spatial maps accordingly. Regarding the calculation of emission changes, we count all the grids in the PRD and NPRD region. We will clarify the calculation method in the revised manuscript.

Line 395: figure 5a is showing the result of PRD, not GD. Please specify.

Response: Accepted and will correct accordingly.

Line 437: "On-road mobile was also a major contributor to NOX emissions in GD (Fig. 5b)." Fig. 5b is showing results of SO2 for NPRD. Please double check.

Response: Thanks for pointing this. We will change the wrong ordinal number "Fig. 5b" to "Fig. 4b" in the revised manuscript.

Line 673: it should be 2.4.4

Response: Accepted. We will correct the number in the revised manuscript.

---

## Author Response (AR1)

**A letter of Response to Reviewers**

We thank the three reviewers for their valuable comments. This manuscript has been improved significantly by addressing these comments. Below are the point-to-point responses to all of the comments. The revision in the manuscript related to the comments was marked in yellow color in the copy of the revised manuscript.

**Response to the Reviewer 1:**

This work describes the long-term emission inventories for seven pollutants throughout Guangdong Province, one of most developed regions in China, which is highly significant. However, the paper still suffers for two detail problems. 1) The single and plural forms for source categories in the article are not very uniform. For example, the emission source –"dust sources" in line 147 is a plural form, but "dust source" is presented in Figure 4. There is an example taken here, which have also been the same problems in the paper. 2) The abscissa of figure 4(d)  $PM_{2.5}$  is diagonal. However, those for the other pollutants are vertical.

**Response**: Thanks for the positive feedback. For the two helpful comments, we accepted and revised on the manuscript.

- Accepted. We unified single and plural forms for source categories. The major categories normalized as power plants, industrial combustion, residential combustion, on-road mobile source, non-road mobile source, dust source, industrial process source, industrial solvent use, non-industrial solvent use, storage and transportation, agricultural source, biomass burning, and other sources. In response to the comment, we revised the description in the manuscript. For example, source categories were revised in Lines 250-253 as "including power plants, industrial combustion, residential combustion, on-road mobile source, non-road mobile source, industrial process source, industrial solvent use, storage and transportation, agricultural solvent use, non-industrial solvent use, storage and transportation, agricultural source, biomass burning, and other sources, were estimated."
- 2) Accepted. The figure 4(d) PM2.5 was replaced with the vertical abscissa in the manuscript (Fig. 4).

(e) VOCs

2008 2009 2010

2007

2006

2012 2013

2011

100%

90% 80%

70%

60%

50%

40%

30%

20%

10%

0%

---

## Author Response (AR2)

**A letter of Response**

**Manuscript ID:** acp-2019-235

**Title:** Evolution of Anthropogenic Air Pollutant Emissions in Guangdong Province, China, from 2006 to 2015

Thank you for accepting our manuscript for publication in ***Atmospheric Chemistry and Physics***. According to the requirement for final revised paper, we refined the manuscript including format of references and figures, and further modified English editing. Thank you very much for all your time.